Neogene amphibians and reptiles (Caudata, Anura, Gekkota, Lacertilia, and Testudines) from the south of Western Siberia, Russia, and Northeastern Kazakhstan

Vasilyan Davit 1 2 3 davit.vasilyan@jurassica.ch
Zazhigin Vladimir S. 4
http://orcid.org/0000-0003-2100-6164 Böhme Madelaine 1 5
1 Department of Geosciences, Eberhard Karls University Tübingen , Tübingen , Germany
2 JURASSICA Museum , Porrentruy , Switzerland
3 Department of Geosciences, University of Fribourg , Fribourg , Switzerland
4 Institute of Geology, Russian Academy of Sciences , Moscow , Russia
5 Senckenberg Center for Human Evolution and Palaeoecology, Eberhard Karls University Tübingen , Tübingen , Germany
Piñeiro Graciela
Electronic publication date: 2017 Mar 23
Publication date: 2017
Volume: 5
Electronic Location ID: e3025
Received 2016 Aug 16; Accepted 2017 Jan 24
Copyright: © 2017 Vasilyan et al.
Copyright year: 2017
Copyright holder: Vasilyan et al.
License: This is an open access article distributed under the terms of the Creative Commons Attribution License, which permits unrestricted use, distribution, reproduction and adaptation in any medium and for any purpose provided that it is properly attributed. For attribution, the original author(s), title, publication source (PeerJ) and either DOI or URL of the article must be cited.
License URL: https://creativecommons.org/licenses/by/4.0/

Keywords: Amphibians, Reptiles, Western Siberia, Neogene, Palaeobiogeography, Palaeoclimate

Funding: DFG BO1550/14-1 SYNTHESYS ES-TAF-2516 The work was supported by DFG (BO1550/14-1) to Madelaine Böhme and SYNTHESYS (ES-TAF-2516) to Davit Vasilyan grants. The funders had no role in study design, data collection and analysis, decision to publish, or preparation of the manuscript.

==============================
Background

The present-day amphibian and reptile fauna of Western Siberia are the least diverse of the Palaearctic Realm, as a consequence of the unfavourable climatic conditions that predominate in this region. The origin and emergence of these herpetofaunal groups are poorly understood. Aside from the better-explored European Neogene localities yielding amphibian and reptile fossil remains, the Neogene herpetofauna of Western Asia is understudied. The few available data need critical reviews and new interpretations, taking into account the more recent records of the European herpetofauna. The comparison of this previous data with that of European fossil records would provide data on palaeobiogeographic affiliations of the region as well as on the origin and emergence of the present-day fauna of Western Siberia. An overview of the earliest occurrences of certain amphibian lineages is still needed. In addition, studies that address such knowledge gaps can be useful for molecular biologists in their calibration of molecular clocks.

Methods and Results

In this study, we considered critically reviewed available data from amphibian and reptile fauna from over 40 Western Siberian, Russian and Northeastern Kazakhstan localities, ranging from the Middle Miocene to Early Pleistocene. Herein, we provided new interpretations that arose from our assessment of the previously published and new data. More than 50 amphibians and reptile taxa were identified belonging to families Hynobiidae, Cryptobranchidae, Salamandridae, Palaeobatrachidae, Bombinatoridae, Pelobatidae, Hylidae, Bufonidae, Ranidae, Gekkonidae, Lacertidae, and Emydidae. Palaeobiogeographic analyses were performed for these groups and palaeoprecipitation values were estimated for 12 localities, using the bioclimatic analysis of herpetofaunal assemblages.

Conclusion

The Neogene assemblage of Western Siberia was found to be dominated by groups of European affinities, such as Palaeobatrachidae, Bombina, Hyla, Bufo bufo, and a small part of this assemblage included Eastern Palaearctic taxa (e.g. Salamandrella, Tylototriton, Bufotes viridis). For several taxa (e.g. Mioproteus, Hyla, Bombina, Rana temporaria), the Western Siberian occurrences represented their most eastern Eurasian records. The most diverse collection of fossil remains was found in the Middle Miocene. Less diversity has been registered towards the Early Pleistocene, potentially due to the progressive cooling of the climate in the Northern Hemisphere. The results of our study showed higher-amplitude changes of precipitation development in Western Siberia from the Early Miocene to the Pliocene, than previously assumed.

Introduction

Western Siberia is a geographic region restricted to the territories of Russia and parts of Northern Kazakhstan. It includes the region between the Ural Mountains in the west, Central Siberian Plateau in the east, and the Kazakh Plain and Altay Mountains, including the Zaisan Lake in the south (Fig. 1). Western Siberia region incorporates the drainage basin of the major Siberian rivers such as the Irtysh and Ob rivers, both flowing into the Kara Sea of the Arctic Ocean. The region is characterised by a highly continental climate, under the influence of the Westerlies (winds). The mean annual precipitation (MAP) is relatively uniform and varies from 400 mm in the north (415 mm at Omsk) to 200 mm in the south (255 mm at Pavlodar). The region has a high relative humidity in summer due to labile convective heating and frequent torrential rainfalls. The mean annual range of temperature reaches 4 °C and more (Omsk: cold month temperature (CMT) −19 °C, warm month temperature (WMT) 20 °C, mean annual temperature (MAT) 0.4 °C; Semipalatinsk: CMT −16 °C, WMT 22 °C, MAT 3.1 °C; Lake Zaisan: CMT up to −27 °C, WMT 23 °C; after Müller & Hennings (2000)). The area is covered by diverse biomes, namely the tundra (‘cold steppe’) and taiga (coniferous forests) biomes, which are replaced by open landscapes in the north (tundra) and in the south (steppe). The region that contains the studied Neogene outcrops is located in the transitional zone between the dry and the more humid temperate biomes, where taiga, forest-steppe and steppe biomes are distributed (Ravkin et al., 2008).

Figure 1 Map of Eurasia (A) showing location of the Western Siberian studied fossil sites (B) (1–38, 58; black—thin outlined circles) as well as localities known from the literature (39–57; white—thick outlined circles).

1, Baikadam; 2, Malyi Kalkaman 2; 3, Malyi Kalkaman 1; 4, Shet-Irgyz 1; 5, Petropavlovsk 1; 6, Znamenka; 7, Pavlodar 1A; 8, Selety 1A; 9, Kedey; 10, Novaya Stanitsa 1A; 11, Borki 1A; 12, Lezhanka 2 A; 13, Cherlak; 14, Pavlodar 1B; 15, Lezhanka 2B; 16, Olkhovka 1A; 17, Olkhovka 1B; 18, Olkhovka 1C; 19, Iskakovka 2 A; 20, Isakovka 1A; 21, Peshniovo 3; 22, Isakovka 1B; 23, Kamyshlovo; 24, Beteke 1B; 25, Pavlodar 2B; 26, Pavlodar 3 A; 27, Lezhanka 1; 28, Andreievka-Speransko; 29, Andreievka 1; 30, Livenka; 31, Beteke 1C; 32, Lebiazhie 1A; 33, Lebiazhie 1B; 34, Podpusk 1; 35, Beteke 2; 36, Beteke 4; 37, Kamen-na-Obi; 38, Razdole; 39, Akespe; 40, Ayakoz; 41, Golubye Peski; 42, Zmei Gorynych; 43, Vympel; 44, Poltinik; 45, Zaezd; 46, Tri Bogatyrja; 47, Kaymanovaja cherepakha; 48, Ryzhaya II; 49, Kentyubek; 50, Ashut; 51, Point ‘Y;’ 52, Sarybulak Svita; 53, Kalmakpai Svita; 54, Karabastuz; 55, Kalmakpai; 56, Petropavlovsk 1/2; 57, Detskaya Zheleznaya Doroga; 58, Shet-Irgyz 2. Map data © 2016 Google and Map data © OpenStreetMap contributors, CC BY-SA.

Due to the strong continental climate, the present-day herpetofauna in the territory of Western Siberia is comparatively far less diverse, represented only by six to 10 amphibian species and seven reptile species (Table 1). It is assumed that the present distribution of amphibians and reptiles in Western Siberia was strongly influenced by Quaternary climatic fluctuation (Ravkin, Bogomolova & Chesnokova, 2010). According to Borkin (1999), the present-day amphibian fauna of Western Siberia belongs to the Siberian region of amphibian distribution in the Palaearctic Realm. According to different authors (e.g. Kuzmin, 1995; AmphibiaWeb, 2016), the region is inhabited by a few amphibians, namely two species of salamanders and four to eight species of anurans, belonging to five genera and five families (Table 1). This is the poorest regional diversity of fauna in the Palaearctic Realm, without any endemic species. Only Salamandrella keyserlingii and Rana amurensis are characteristic of the territory, but they are widely distributed and are also found in smaller areas in the neighbouring regions (Borkin, 1999). The Western Siberian reptile fauna listing includes few species: Natrix natrix, Elaphe dione, Vipera berus, Vipera renardi, Gloydius halys, Zootoca vivipara, Lacerta agilis, Eremias arguta (Ananjeva et al., 2006; Ravkin, Bogomolova & Chesnokova, 2010).

Table 1 Recent herpetofauna of southwestern part of Siberia (Ob and Irtysh River drainages) according to different authors.

Taxa	Reference	
1	2	3	4	5	
Caudata	Salamandrella keyserlingii	+	+	+	+		
Lissotriton vulgaris	is	+	−	+		
Anura	Rana arvalis	+	+	+	+		
Rana amurensis	+	+	+	+		
Rana temporaria	+	−	+	+		
Pelophylax ridibundus	is	−	+	is		
Bufotes viridis	is	+	+	is		
Bufotes variabilis	−	−	−	+		
Bufo bufo	+	+	+	+		
Bufo gargarizans	−	?	−	is		
Lacertoidea	Lacerta agilis					+	
Zootoca vivipara					+	
Serpentes	Elaphe dione					+	
Natrix natrix					+	
Vipera berus					+	
Vipera renardi					+	
Gloydius halys					+	
Notes:

Recent herpetofauna of southwestern part of Siberia (Ob and Irtysh River drainages) according to different authors. Reference: 1, Kuzmin (1995); 2, Borkin (1999); 3, Ravkin, Bogomolova & Chesnokova (2010); 4, AmphibiaWeb (2016); 5, Ananjeva et al. (2006). is, insular occurrence.

Geology and stratigraphy

The Neogene sediments in Western Siberia have a wide distribution. Over many decades, through systematic palaeontological studies and research in the Neogene and Quaternary sediments of this area, rich fossil deposits of molluscan and small and large mammalian faunas have been discovered (e.g. Zykin, 1979; Zykin & Zazhigin, 2008; Zykin, 2012). Based on the studies of the small fossil mammals, the Neogene stratigraphy of the area is complemented with biochronologic data. Continental sedimentation in the western part of the Siberian Plain began in the Oligocene, after regression of the Turgai Strait in the late Eocene, and continued until the Quaternary period (e.g. Chkhikvadze, 1984, 1989; Tleuberdina et al., 1993; Malakhov, 2005). The sedimentary basin is surrounded by the Ural Mountains in the west, the Central Kazakh Steppe and Altai-Sayan Mountains in the south, and the western margin of the Siberian Plateau in the east. The surrounding regions deliver clastic material to the basin. Some researchers include the Zaisan Basin, located to the west of the Altai-Sayan Mountains in this territory (Borisov, 1963). The Neogene sediments are represented by lacustrine, fluvial, alluvial, and other continental depositions, overlying marine Eocene sediments. The thickest section (300 m) of the Neogene and early Quaternary sediments occurs in the Omsk Basin. Neogene strata outcrops are mainly found in the interfluves of the Irtysh and Ishim rivers (Gnibitenko, 2006; Zykin, 2012). All these sediments are terrestrial (fluvial and alluvial facies) and have produced rich fossil layers of vertebrate fauna (Zykin, 2012). The vertebrate-bearing Neogene sediments are found in several areas along the Irtysh River and its tributaries—Petropavlovsk–Ishim (e.g. Petropavlovsk 1, Biteke 1A), Omsk (e.g. Novaya Stanitsa 1, Cherlak), Pavlodar (e.g. Pavlodar, Baikadam) and the Novosibirsk areas (e.g. Kamen-na-Obi) (Fig. 1). Detailed geological descriptions of the stratigraphic sections and fossil localities are summarised in Zykin (1979), Zykin & Zazhigin (2004), Gnibitenko (2006), Zykin (2012).

The stratigraphic subdivision is based mainly on the Russian concept of svitas. A svita has lithologic, biochronologic, and genetic (sedimentologic) significance and has no precise equivalent in western stratigraphic theory and terminology (Lucas et al., 2012). The stratigraphy of Neogene sediments in Western Siberia is supported by magnetostratigraphic investigations (e.g. Gnibitenko, 2006; Gnibidenko et al., 2011), in which the recovered polarity signals are combined with biochronologic data and correlated to the geomagnetic polarity time scale (Fejfar et al., 1997; Vangengeim, Pevzner & Tesakov, 2005; Zykin, Zykina & Zazhigin, 2007). The biozonation is based on fast-evolving lineages of small mammals, mainly jerboas (Dipodidae), hamsters (Cricetidae) and voles (Arvicolidae). Owing to these bio-magnetostratigraphic data, the mean temporal resolution of the late Neogene faunal record from the Ob–Irtysh Interfluve is estimated to be approximately 200 kyr (Fig. 2; Table S1; Data S2). The main sections of these vertebrate fossil localities are referred to certain svitas (e.g. Kalkaman, Pavlodar, Irtysh Svitas), however, the stratigraphic assignment of three localities Olkhovka 1A, 1B, 1C to svitas is not available (Fig. 2; Table S1). No fossils are available in the initial deposits of the early Late Miocene.

Figure 2 Compiled stratigraphy of the Middle Miocene–Early Pleistocene studied localities in Western Siberia grouped in the svitas and relative to their geographic positions.

The localities without assignment into a certain svita are given in coloured frames according to age. In the right column, the small mammalian biochronologic data (species or lineages) used for age estimations are given; the numbers accompanying the species refer the locality numbers. The arrows to the left from the small mammal taxa indicate its/their first appearance. Abbreviations: klm, Kalkaman; ish, Ishim; pv, Pavlodar; kd, Kedey; nst, Novaya Stanitsa; rt, Rytov; is, Isakov; psh, Peshnev; krt, Krutogor; bt, Betekey; liv, Levetin; irt, Irtysh; kar, Karagash.

State-of-the-art palaeoherpetological studies in Western Siberia

The fossil record of amphibians and reptiles in Western Siberia, including the Zaisan Basin record, remain largely unknown. There are very few works devoted to the studies of the Western Siberian late Paleogene and Neogene herpetofaunal assemblages (e.g. Chkhikvadze, 1984, 1989; Tleuberdina et al., 1993; Malakhov, 2005). The vast majority of data on fossil amphibians and reptiles are represented as short notes or are mentioned in faunal lists (e.g. Bendukidze & Chkhikvadze, 1976; Chkhikvadze, 1985; Malakhov, 2005). In this present contribution, we analysed the available data from specimens described below and from new generated data as well.

The earliest report on Neogene fossil amphibians was compiled by Iskakova (1969), wherein she described amphibian faunas from two Priirtyshian localities, Gusiniy Perelet and Karashigar. Gusiniy Perelet is a well-renovated Late Miocene vertebrate fossil locality, situated on the riverbank of the Irtysh River, within the town of Pavlodar. The sedimentary sequence in this locality contains layers of different ages from the late Late Miocene until the late Early Pliocene. Three localities (also ‘horizons’) within the town of Pavlodar (Pavlodar 1A, 1B, 3B) are grouped into several svitas and can be distinguished from the Gusiniy Perelet vertebrate locality. The fossil content of the Gusiniy Perelet locality comes from the lower horizon—Pavlodar 1A. Iskakova (1969) described an amphibian fauna from this layer.

The age of the Karashigar locality is unclear. In a study by Tleuberdina, Kozhamkulova & Kondratenko (1989), this locality has been estimated to date back to the Late Oligocene; however, Lychev (1990) placed it in the Middle Miocene, Kalkaman Svita (the list of the small mammal fauna; see Data S2). The amphibian taxa described by Iskakova (1969) in the Priirtyshian localities (Bombina cf. bombina, Pelobates cf. fuscus, Bufo cf. viridis, Bufo cf. bufo, Rana cf. ridibunda, Rana cf. temporaria) were identified based mainly on the vertebrae (cervical, dorsal and sacral) morphology, which is not diagnostic in frogs at that taxonomic level. Chkhikvadze (1984) restudied the material from the Pavlodar 1A (=Gusiniy Perelet) locality and identified Bufo cf. raddei, Bufo sp., Pelophylax cf. ridibundus, Eremias sp., and Coluber sp. In this study, we did not, however, assess the material from the above-mentioned works in order to verify Chkhikvadze (1984) taxonomic identifications. Our sample from this locality (Pavlodar 1A) (Table S1) did not reveal any element listed in these earlier studies (Chkhikvadze, 1984; Iskakova, 1969).

Chkhikvadze (1984) summarised all known fossil amphibians and reptiles from the former Union of Soviet Socialist Republics (USSR), including those from Western Siberia. Accurate descriptions are not yet available for many of these species. The Middle Miocene Kalkaman locality (Tleuberdina, 1993), presently known as Malyi Kalkaman 1 (Zykin, 2012), has provided a diverse record of fossil herpetofauna. The fossil record of this locality was partially restudied and amended by us, which included the collection of new material.

Over the last decade, fresh attempts have been made to study the herpetofauna from the Western Siberian localities (Malakhov, 2003, 2004, 2005, 2009). In the resultant works, undescribed material from several Neogene localities of Kazakhstan were summarised, revised, and studied, thereby providing critical overviews. In spite of the advances of the recent years, however, the Neogene herpetofauna from Western Asia remains largely unknown, with available fossil material continuing to be insufficiently studied. The main goals of the present study were, therefore, to assess the descriptions and taxonomic classifications of the new amphibian and reptile fossil material collected by Vladimir Zazhigin (co-author), as well as already published data so as to provide a comprehensive faunistic analysis and palaeobiogeographic and environmental interpretations. To avoid confusion around the names used by different authors in the Russian literature to describe the localities, we have provided all known names for these studied fossil localities.

Materials and Methods

The new materials used in the present study were collected by V. Zazhigin (co-author) using the screen-washing technique during his long-term excavations in different Western Siberian localities from the 1960s to 2008. These localities outcrop along the riverbanks of the Irtysh, Ishim, and Ob rivers. This fossil material is stored in the Institute of Geology, Russian Academy of Sciences under the collection numbers: GIN 950/2001 (Baikadam), GIN 1107/1001 (Malyi Kalkaman 1), GIN 1107/2001 (Malyi Kalkaman 2), GIN 1106/1001 (Shet Irgyz 1), GIN 952/1001 (Petropavlovsk 1), GIN 1109/1001 (Znamenka), GIN 640/5001 (Pavlodar 1A), GIN 951/1001 (Selety 1A), GIN 951/2001 (Kedey), GIN 948/2001 (Novaya Stanitsa 1A), GIN 1115/1001 (Borki 1A), GIN 1110/2001 (Cherlak), GIN 945/2001 (Beteke 1A), GIN 640/6001 (Pavlodar 1B), GIN 1130/1001 (Lezhanka 2A), GIN 1130/2001 (Lezhanka 2B), GIN 1111/1001 (Olkhovka 1A), GIN 1111/2001 (Olkhovka 1B), GIN 1111/3001 (Olkhovka 1C), GIN 1118/3001 (Peshniovo 3), GIN 1131/2001 (Isakovka 2), GIN 1131/1001 (Isakovka 1A), GIN 1131/3001 (Isakovka 1B), GIN 1117/1001 (Kamyshlovo), GIN 945/2001 (Beteke 1B), GIN 945/3001 (Beteke 1C), GIN 1112/1001 (Andreievka–Speranskoe), GIN 1108/2001 (Pavlodar 2B), GIN 1112/2001 (Andreievka 1), GIN 1129/2001 (Livenka), GIN 1129/1001 (Lezhanka 1), GIN 1108/3001 (Pavlodar 3A), GIN 950/3001 (Lebiazhie 1A), GIN 950/4001 (Lebiazhie 1B), GIN 950/5001 (Podpusk 1), GIN 945/60001 (Beteke 2), GIN 946/2001 (Kamen-na-Obi), GIN 945/8001 (Beteke 4), GIN 664/2001 (Razdole).

Various groups of amphibians and reptiles are represented in the available material. A report of part of this material, i.e. of the anguine lizards, has been published in a separate paper (e.g. Vasilyan, Böhme & Klembara, 2016). The present study included an assessment of the materials collected from four fossil sites in Kazakhstan: Akyspe (also known as Agyspe), Aral Horizon, leg. by Bendukidze in 1977; Kentyubek, Turgai Basin; Ryzhaya II (Ryzhaya Sopka), Zaisan Svita, Zaisan Basin, leg. in 1970; Ayakoz (known also as Ayaguz), Zaisan Basin, leg. in 1970–1971; Petropavlovsk 1/2,1 leg. 1972 (Table S1). In addition, the few available data from the literature were included in this study (after critical revision) to amend the record of herpetofaunal assemblages of some localities as well as to reassign and revise the stratigraphic position of these localities using biochronologic information of small and large mammalian fauna (see full list in Datas S2 and S3).

The photographs of the fossil material were taken using a digital microscope, Leica DVM5000 (Tübingen, Germany) and inspected with a scanning electron microscope, FEI Inspect S (Madrid, Spain). The figures and tables were produced using Adobe Photoshop and Illustrator programs. The osteological nomenclature of this study followed that of Vasilyan et al. (2013) for the salamander remains, that of Sanchíz (1998a) for frogs, that of Daza, Aurich & Bauer (2011) and Daza & Bauer (2010) was used for Gekkota, and the lepidosaurian terminology of Evans (2008).

Based on the herpetofaunal assemblages, the palaeoprecipitation values for the fossil localities were estimated using the method of bioclimatic analysis of the ecophysiologic groups of amphibian and reptile taxa (Böhme et al., 2006). For the localities considered to be ‘poor’ in amphibian and reptile taxa, the range-through approach (Barry et al., 2002) was used, in which the faunas of two or more localities with age differences less than 100 kyr and/or belonging to a single stratigraphic unit—svita, were considered as one. The taxa that were added to the herpetofaunal assemblage using the range-through approach are indicated in grey in Table S4.

Results

Systematic palaeontology

Class Amphibia Gray, 1825

Order Caudata Scopoli, 1777

Family Hynobiidae Cope, 1859

Genus Salamandrella Dybowski, 1870

Salamandrella sp.

(Figs. 3D–3G)

Figure 3 Salamander remains from Western Siberian localities.

(A–C), Cryptobranchidae indet. from the loc. Gusiny Perelet, unnr. PIN specimens; (A) fragmentary right dentary, natural cross-section; (B) the same dentary, in lingual view; (C) a jaw fragment, lingual view; (D–G) Salamandrella sp., Lezhanka 2 A, GIN 1130/1001-AM01, trunk vertebra; (H–S) Mioproteus sp.; (H–L) loc. Ayakoz, trunk vertebra, GNM unnr. specimen; (M–O) trunk vertebra, Borki 1A, GIN 1115/1001-AM01; (P, Q) right premaxilla, Malyi Kalkaman 2, GIN 1107/2001-AM01; (R, S) left premaxilla loc. Grytsiv (Ukraine), unnr. MNMHK specimen; (T–X) trunk vertebrae of aff. Chelotriton sp., loc. Ayakoz, GNM unnr. specimen; (Y) Chelotriton sp. from Malyi Kalkaman 2, GNM unnr. specimen; (D, H, M, P, R, T, Y) dorsal view; (E, I, N, Q, S, U) ventral view; (F, J, O, V) lateral view; (G, K, W) anterior view; (L, X) posterior view. Scale bars: A–C = 5 mm; D–G = 0.5 mm; H–Y = 1 mm.

Localities and material examined

Malyi Kalkaman 1, GIN 1107/1001-AM12, one right femur; Selety 1A, GIN 951/1001-AM01–AM03, three trunk and GIN 951/1001-AM04, one caudal vertebra; GIN 951/1001-AM05, one distal end of bone (humerus?); Novaya Stanitsa 1A, GIN 948/2001-AM01–AM11, 11 trunk vertebrae; Lezhanka 2A, GIN 1130/1001-AM01–AM26, 26 trunk and GIN 1130/1001-AM27–AM28, two caudal vertebrae; Cherlak, GIN 1110/2001-AM01–AM12, 12 trunk vertebrae; Lezhanka 2B, GIN 1130/2001-AM01, one trunk vertebra, GIN 1130/2001-AM02, one extremity bone; Olkhovka 1B, GIN 1111/2001-AM01, one trunk vertebra; Iskakovka 2A, GIN 1131/2001-AM01, one trunk vertebra; Andreievka–Speransko, GIN 1112/1001-AM01, one trunk vertebra; Lezhanka 1, GIN 1129/1001-AM01–AM02, two trunk and GIN 1129/1001-AM03, one caudal vertebrae; Beteke 1C, GIN 945/3001-AM01–AM02, two trunk vertebrae.

Description and comments

The vertebrae have an elongated to nearly slender form. The vertebral centrum is amphicoelous. The basapophyses at the vertebral centrum are either absent or are present in the form of a small protuberance at the laterodorsal corners of the anterior portion of the vertebral centrum (Fig. 3G). A pair of subcentral foramina is situated at the basis of the transverse processes. The neural arch is tall in lateral view (Fig. 3F) and relatively broad in dorsal view (Fig. 3D). The posterior edge of the pterygapophysis is bifurcated. Sometimes the neural spine is present but in general the dorsal surface of the neural arch is flat. The pre- and postzygapophyses have an elongated oval shape. In anterior view, the neural canal has an outline of a regular pentagon. The transverse process is unicapitate (Figs. 3D and 3G). The anterior and posterior alar processes are absent. The vertebrae can be assigned to the family Hynobiidae based on: (1) the small size and their amphicoelous centrum with circular articular surfaces; (2) the lack of or being weakly pronounced basapophyses; (3) the lack of neural spine; (4) the notch on the posterior margin of neural arch; (5) the fused rib-bearers; and (6) the intervertebrally exiting spinal nerve in both trunk and caudal vertebrae (e.g. Edwards, 1976; Venczel, 1999a, 1999b). Further, characteristic features can be observed on the vertebrae of representatives of the genus Salamandrella, namely the absence of the subcentral foramen and the concave anterior margin of the neural arch that reaches the middle part of the prezygapophyseal articular facets (Venczel, 1999b; Ratnikov & Litvinchuk, 2009; Syromyatnikova, 2014) (Figs. 3D–3G). The detailed description of hynobiid material from the Western Siberian localities and comparison with recent and fossil hynobiids is provided in a forthcoming paper.

Family Cryptobranchidae Fitzinger, 1826

Cryptobranchidae indet.

(Figs. 3A–3C)

Localities and material examined

Pavlodar 1A (=Gusiniy Perelet), one fragmentary right dentary and two fragments of jaw bones, for details about the stratigraphic allocation see section ‘Cryptobranchidae’, unnr. PIN specimen.

Description and comments

Among the fragments, a posterodorsal portion of a large right dentary, 27 mm in length, is present. In lingual view, the pars dentalis is composed entirely of dental lamina and the subdental lamina is present, but reduced. The pars dentalis possesses 30 pedicels of pleurodont teeth. The subdental shelf inclines slightly ventrally. The lamina horizontalis is prominent. The corpus dentalis above the Meckelian groove has a concave surface. Ventrally, this surface possesses a ridge running parallel to the lamina horzontalis. The cross section of the dentary shows a relatively low portion of cancellous bone and a dominance of compact bone. The size of the bones, the form and structure of the pars dentalis and the cross section of the bone are characteristic of giant salamanders (Vasilyan et al., 2013).

Family Proteidae Gray, 1825

Genus Mioproteus Estes & Darevsky, 1977

Mioproteus sp.

(Figs. 3H–3S)

Localities and material examined

Ryzhaya II (known also Ryzhaya Sopka), GNM unnr. specimen, two trunk vertebrae; Malyi Kalkaman 2, GIN 1107/2001-AM01, one right premaxilla; Borki 1A, GIN 1115/1001-AM01, one trunk vertebra; Ayakoz, GNM unnr. specimen, one trunk vertebra; Akespe, unnr. HC specimens, three vertebrae; Petropavlovsk 1/2, GNM unnr. specimen, 22 vertebrae.

Description

The preserved left premaxilla is fragmentary (Figs. 3P and 3Q) and the posterior process is broken off. In ventral view, the bone has a rough surface. The pars dentalis of the premaxilla is located on the anterior side of the bone. The crowns of pleurodont teeth are missing and only their pedicellar portions are preserved. The bone surface is slightly rough in dorsal view. The lamelliform anterolateral ridge of the posterior process is high at the middle part of the bone. The amphicoelous vertebrae are flat and wide. The centrum is dumb-bell in shape and narrows to the middle region. The basapophyses, if present, are small and weakly developed. Two subcentral foramina are present at the central part of the vertebral centrum. In lateral view, the vertebra is low; the anterior and posterior zygapophyseal crests are pointed, forming the dorsal border of the deep depressions anteriorly and posteriorly to the transverse process. The middle part of the neural arch is lower than its cranial and caudal margins. The posterior edge of the neural arch is forked (Fig. 3H) (not visible at Fig. 3M). The neural spine extends as far as the preserved anterior margins of the neural arch, whereas posteriorly, it terminates before the posterior margin of the neural arch. The preserved right pre- and postzygapophyseal articular facets are ellipsoid.

Comparison and comments

A direct comparison with Mioproteus specimens from previous reports was not possible due to the extremely scarce description of the skull elements attributed to this taxon (e.g. Estes & Darevsky, 1977; Miklas, 2002). We therefore used the material of Mioproteus sp. from the Grytsiv locality (Ukraine, earliest Late Miocene) (Figs. 3R and 3S) for the taxonomic identification of the fossil premaxilla from Malyi Kalkaman 2 (Figs. 3P and 3Q). Our comparison founds no differences in the premaxilla morphology between the Kazakhstan and Ukrainian Mioproteus sp. The vertebrae from the Borki 1A and Ayakoz localities can be easily assigned to the genus Mioproteus based on following characters: (1) robust vertebra with an amphicoelous centrum; (2) a tall cranial margin of the neural arch; (3) the presence of the basapophyses; (4) a distinct wide depression at the anterior base of the transverse process; (5) intervertebrally exiting spinal nerves; and (6) a forked neural spine (Edwards, 1976; Estes & Darevsky, 1977; Ivanov, 2008).

Family Salamandridae Goldfuss, 1820

Subfamily Pleurodelinae Tschudi, 1838

Genus Chelotriton Pomel, 1853

Chelotriton sp.

(Figs. 3T–3Y)

Localities and material examined

Malyi Kalkaman 1, GNM unnr. specimen, one trunk vertebra; Ayakoz, GNM unnr. specimen, one trunk vertebra.

Description

The single fragmentary trunk vertebra of Chelotriton from the Malyi Kalkaman 1 locality has been scantily described (Tleuberdina et al., 1993, 133–134). The centrum of the vertebra is ophistocoelous and dorsally curved. Both the posterior one-third of the vertebra and cotyle are broken. The condyle is dorsoventrally slightly compressed and oval in shape. The middle part of the ventral surface of the centrum bears a pair of the foramina subcentrale. The ventral bases of both transverse processes are pierced by a foramen (potentially the ventral foramen for the spinal nerve).

The neural spine is tall, long, and almost equal in length to the vertebral centrum. The dorsal surface of the neural spine has the form of an elongated isosceles triangle and it is covered by a distinct pustular sculpture. The anterior margin of the neural spine is concave in outline. The posterior half of the spine is wider than the anterior one (Fig. 3Y). In anterior view, the neural arch and the neural canal have a triangular form. The roof of the neural canal is flat, on both sides of the neural spine.

The pre- and postzygapophyses are damaged. The anterior portion of the left postzygapophysis is present and it shows a horizontal surface. The anterior bases of both prezygapophyses at the contact with the centrum possess small subprezygapophyseal foramina. Behind the left prezygapophysis, the accessory alar process exhibits a marked step (Fig. 3Y), projects posteroventrally and connects caudally with the anterior alar process. The contact point of the accessory and anterior alar processes probably corresponds to the base of the parapophysis. Both transverse processes are broken, but the bases are preserved. Apparently, two rounded upper and lower prominences, seen in left lateral view, correspond to the dia- and parapophysis. The parapophysis is located anteriorly and dorsally to the level of the diapophysis; thus, the transverse process becomes a bent projection. The arterial canal runs behind the base of the transverse process. Anteriorly, its dorsal and ventral walls are built by the accessory and anterior alar processes.

The vertebra from the Ayakoz locality (Figs. 3T–3X) is fragmentary, its neural arch and left transverse process are lost, the centrum is compact, short and wide, and it possesses an elliptical central foramen. The diapophysis of the preserved right transverse process is broken, but it can be assumed that the dia- and parapophysis were separated from each other. The accessory alar process runs from the prezygapophysis to the dorsal edge of the diapophysis. The posterior and anterior alar processes run from the cotyle and condyle straight along the transverse process to the parapophysis. This morphology is characteristic of the first trunk vertebrae.

Comparison and comments

This vertebra was previously described by Tleuberdina et al. (1993). Here, we have assigned this specimen to the genus Chelotriton owing to the presence of a triangular and well-sculptured plate on the top of the neurapophysis. This character, however, is not a unique feature of Chelotriton and is also seen in other salamanders, e.g. recent species of Tylototriton and Echinotriton, and in Cynops pyrrhogaster, Lissotriton boscai (unnr. GPIT specimen), Paramesotriton (MNCN 23557, 13645), as well as the fossil taxa Archaeotriton (Böhme, 1998), aff. Tylototriton sp. (Baikadam locality, this paper), Carpathotriton (Venczel, 2008). The vertebra from the Malyi Kalkaman 1 resembles the species of Chelotriton, Paramesotriton, Tylototriton, Echinotriton, Cynops pyrrhogaster, and Carpathotriton in their mutual presence of a subprezygapophyseal foramen. The vertebra can, however, be justified as Chelotriton sp. and distinguished from other salamanders by: (1) its longer length (vs. Echinotriton, Cynops, and Carpathotriton); (2) a longer neural spine with a rugose sculptured and triangular dorsal surface (vs. aff. Tylototriton sp., Baikadam locality, this paper); and (3) a well-pronounced accessory alar process (vs. Tylototriton).

The fragmentary vertebra from the Ayakoz locality can be assigned also to this group because of the presence of massive rib-bearers and large dimensions (Ivanov, 2008). Its vertebra is identical to that of vertebra of Chelotriton sp. type II described from the Mokrá-Western Quarry, 2/2003 Reptile Joint (Early Miocene, Czech Republic) (Ivanov, 2008).

The abundant European Cenozoic record of the genus Chelotriton, however, showed that vertebral morphology is insufficient for taxonomic identification as Chelotriton (Böhme, 2008). This genus has an unknown higher diversity, which can be uncovered by the study of complete skeletons of those species. We hence classified the vertebrae from studied localities as aff. Chelotriton sp.

Genus Tylototriton Anderson, 1871

aff. Tylototriton sp.

(Figs. 4A–4K)

Figure 4 Trunk vertebrae of fossil aff. Tylototriton (A–K) and recent Tylototriton, Echinotriton and Cynops (L–AE).

(A–E) aff. Tylototriton sp., locality Ayakoz, GNM unnr. specimen; (F–K) GIN 950/2001-AM14 and GIN 950/2001-AM01, loc. Baikadam; (L–P) Tylototriton verrucosus, GPIT unnr. specimen; (Q–U) Tylototriton shanjing, GPIT unnr. specimen; (V–Z) Echinotriton andersoni, GPIT unnr. specimen; (AA–AE) Cynops pyrrhogaster, GPIT unnr. specimen; (A, F, G, L, Q, V, AA) lateral view; (B, H, M, R, W, AB) dorsal view; (C, I, N, S, X, AC) ventral view; (D, J, O, T, Y, AD) anterior view; (E, K, P, U, Z, AE) posterior view. Scale bars = 2 mm.

Locality and material examined

Baikadam, GIN 950/2001-AM01, -A14–A17, five trunk vertebrae; Ayakoz, GNM unnr. specimen, two trunk vertebrae.

Description

All preserved vertebrae are opisthocoelous. The condyle and cotyle are dorsoventrally compressed. The vertebrae are slender, slightly narrow, and high. The neural canal is round, but in anterior view, the ventral margin of the neural canal is flat. The same occurs with the dorsal wall of the vertebral centrum. The centrum is dorsally curved in lateral view (Figs. 4A, 4F and 4G). The neural spine was most probably high but does not reach the level of the pustular region. The neural spine begins behind the cranial margin of the neural arch. The neural arch is tilted dorsally and does not extend beyond the posterior edge of the postzygapophysis. The dorsal plate of the neural spine is short, poorly developed, and covered with rugosities. It has the form of an isosceles triangle. Due to the concave shape of the posterior margin of the caudal border, we suggest that the neural spine was probably bifurcated. The length of the neural spine, without the sculptured structure, is the same in all preserved vertebrae and corresponds nearly to almost half of the entire vertebral length (Figs. 4A, 4F and 4G).

The pre- and postzygapophyses are horizontal and almost at the same level (e.g. Fig. 4A). The pre- and postzygapophyseal articular facets are oval in shape. Small subprezygapophyseal foramina are present at the level of the connection between the anterior bases of both prezygapophyses with the vertebral centrum. The posterolaterally directed transverse process is horizontally flattened and displays a bicapitate articulation surface with the rip. The diapophysis and parapophysis are separated, with the former being smaller than the latter. A low and moderately deep notch is developed at the posterior edge of the neural arch. The transverse process has an anterior (accessory alar process) and posterior laminar edges (i.e. the posterior alar process and dorsal lamina). The straight, posteroventrally directed accessory alar process connects the prezygapophysis caudally with the base of the parapophysis (e.g. Fig. 4F). The dorsal lamina starts from the diapophysis and extends to the postzygapophyses, whereas the lamelliform posterior alar process starts at the parapophysis and terminates directly before the cotyle. Subparallel to the accessory alar process, a thin anterior alar process runs along the cranial half of the centrum. Behind and in front of the transverse process two ‘cavities’ (a shallow anterior and a deep posterior) are present. These ‘cavities’ are connected by a canal (possibly an arterial canal), that runs through the transverse process. In ventral view, the vertebral centrum does not possess a ventral keel. The centrum is flattened and nearly plane in the middle portion. Its surface is rough and pierced by numerous foramina. Two large subcentral foramina are located at the posterior corner between the centrum and transverse process (Figs. 4C and 4L).

Comparison and comments

The vertebrae resemble the morphology of pleurodeline salamanders Echinotriton, Tylototriton, Cynops, Chelotriton, Paramesotriton and Tylototriton, and Carpathotriton in characteristics such as: (1) the presence of rugosities on the neural arch; (2) the connection of the prezygapophysis and parapophysis with the accessory alar process, except in Carpathotriton, Cynops, and cf. Tylototriton sp. from Möhren 13 (Böhme, 2010; p. 11, Fig. 6F), where this process connects prezygapophysis with diapophysis; (3) a moderately developed posterior ‘cavity’ behind the transverse process; and (4) the presence of subprezygapophyseal foramen (for collection references see subsection ‘Comparison’ of Chelotriton sp. in this report). In terms of the general morphology, the vertebrae mainly resemble the genus Tylototriton and differ from the compared genera in having: (1) a low, elongate, narrow and lesser flattened vertebrae; (2) a weakly developed pustular structure of the neural arch (similar character as seen in Paramesotriton); (3) a low and long neural spine without the sculptured structure; (4) a dorsoventrally compressed cotyle and condyle; (5) a deep posterior ‘cavity’ behind the transverse process, and an extended dorsal lamina and posterior alar process; (6) a low and shallow posterior notch of the neural arch; and (7) in having an accessory alar process that reaches the parapophysis, which differs from specimens of the genus Cynops wherein it reaches the diapophysis. The Siberian Tylototriton differs from the European Oligocene cf. Tylototriton (see Böhme, 2010; p. 11, Fig. 6F) by having: (1) a ventrally deflected accessory alar process that terminates ventrally to the parapophysis; (2) a shorter and lower neural spine; and (3) a shorter dorsal plate of the neural spine.

Figure 6 Fossil frogs from Western Siberia.

(A–L, P, Q, X, AA, AB, A, AF) Ilia; (A–C) Bombina cf. bombina, Selety 1A, GIN 951/1001-AM06; (D, E) Bombina sp., Cherlak, GIN 1110/2001-AM13; (G–I) Pelobates sp., Selety 1A, GIN 951/1001-AM07; (J–L) Hyla gr. H. savignyi, Lezhanka 2 A, GIN 1130/1001-AM29; (P, Q) Bufo bufo, Olkhovka 1B, GIN 11 11/2001-AM03; (X) Bufotes cf. viridis, Pavlodar 1A, GIN 640/5001-AM01; (AA, AB) Pelophylax sp., Lezhanka 1, GIN 1129/1001-AM05; (AE, AF) Rana arvalis, Malyi Kalkaman 1, GIN 1107/1001-AM10; (A, D, G, J, P, U, AA, AE) in lateral view; (B, E, H, K, Q, AB) in proximal view; (C, F, I, L) in medial view; (M–O, R–T, Y, Z, AC, AD, AG, AH) scapulae of frogs; (M–O) Hyla gr. H. savignyi from Lezhanka 2 A, GIN 1130/1001-AM33; (R–T) Bufo bufo, Olkhovka 1C, GIN 1111/3001-AM01; (Y, Z) Bufotes cf. viridis, Pavlodar 1A, GIN 640/5001-AM63; (AC, AD) Pelophylax sp., Lezhanka 1, GIN 1129/1001-AM07; (AG, AH) Rana temporaria, Malyi Kalkaman 1, GIN 1107/1001-AM01; (M, R, Y, AC, AG) dorsal view; (N, S, Z, AD, AH) ventral view; (O, T) posterior view; (U, V) trunk vertebra of Bufo bufo, Olkhovka 1C, GIN 1111/3001-AM02; (U) anterior view; (V) lateral view; (W) urostyle of Bufo bufo, Olkhovka 1C, GIN 1111/3001-AM03, dorsal view. The arrows show the position of the angular fossa. Scale bars: A–Q, AA–AD, AG, AH = 1 mm, R–Z, AE, AF = 2 mm.

Taking into account the above-mentioned differences, we suggest that the described vertebrae should be assigned to a new pleurodeline salamander genus that shows affinities with the genus Tylototriton. However, we do not consider it reasonable to describe a new form unless cranial material of this salamander is available.

Order Anura Fischer von Waldheim, 1813

Family Palaeobatrachidae (Cope, 1865)

Palaeobatrachidae indet.

(Figs. 5A–5D)

Figure 5 Palaeobatrichid sphenethmoids.

(A–D) Palaeobatrachidae indet., Novaya Stanitsa 1A, GIN 948/2001-AM12; (E–H) Palaeobatrachus sp. from Grytsiv (Ukraine), unnr. NMNHK specimen; (A, E) ventral view; (B, F) dorsal view; (C, G) anterior view; (D, H) lateral view. Abbreviations: ao, antrum olfactorium; alo, antrum pro lobo olfactorio; is, incisura semielliptical; ff, frontoparietal facet; lp, lateral processes; ls, lamina supraorbitalis; nf, nasal facet; onf, orbitonasal foramina; olf, olfactory foramina; pf, parasphenoid facet. Scale bars = 1 mm.

Locality and material examined

Novaya Stanitsa 1A, GIN 948/2001-AM12, one sphenethmoid.

Description

This specimen is represented by a very robust sphenethmoid that lacks the posterior region. The two anterior cavities corresponding to the antrum olfactorium are anteroposteriorly shallow. The posterior cavity, antrum pro lobo olfactorio, is deep and narrow (Figs. 5A and 5B). The olfactory foramen is larger than the orbitonasal foramen (Fig. 5C). The processus rostralis is elongated and projects anteriorly. Anteriorly, dorsal face of the bone, two sharply marked crescentic depressions (nasal facets) correspond to the contacts with the nasal bones (Fig. 5A). In dorsal view, the frontoparietal facet (contacting with the frontoparietal cranial bones) shows a slightly striated surface. The lateral processes protrude laterally. The lamina supraorbitalis is well developed. The most anterior part of the incisura semielliptical is preserved on the specimen. The remaining part of this structure demonstrates that it approaches cranially to the anterior border of the bone. The ventral face of the sphenethmoid possesses a narrow and long depression corresponding to the contact area with the cultriform process of the parasphenoid (the parasphenoid facet) (Fig. 5B).

Comparison and comments

The bone has strong similarities to that of palaeobatrachids in having: (1) a long sphenethmoid with a frontoparietal fenestra corresponding to more than half of the bone length; (2) the articulation area of the parasphenoid delimited by two parallel ridges, in ventral view; and (3) a very short septum nasi and lateral process (Vergnaud-Grazzini & Młynarski, 1969; Sanchíz & Młynarski, 1979). The palaeobatrachid from the Novaya Stanitsa 1A locality exhibits all these characters aside from the short septum nasi, which is longer in the fossil bone. We presume that the frontoparietal fenestra was longer more than half of the sphenethmoid length because the overall length of the frontoparietal and nasal facets has similar proportions as these seen in other palaeobatrachids. Furthermore, according to Venczel, Codrea & Fărcaş (2012), the sphenethmoidal ossification forms the anterior margin of frontoparietal fontanelle in palaeobatrachid frogs (Palaeobatrachus + Albionbatrachus), which can also be observed in the studied specimen.

Family Bombinatoridae Gray, 1825

Genus Bombina Oken, 1816

Bombina sp./Bombina cf. bombina (Linnaeus, 1761)

(Figs. 6A–6F)

Localities and material examined

Malyi Kalkaman 2, GIN 1107/2001-AM02, one ilium; Selety 1A, GIN 1107/2001-AM06, one ilium; Cherlak, GIN 1107/2001-AM06, one ilium.

Description

The bone description is based on the ilium from the Selety 1A locality, since the specimens from the Malyi Kalkaman 2 and Cherkal localities are greatly damaged. In lateral view, the iliac shaft is almost straight and lacks the dorsal crest. The tuber superior is a weakly pronounced tubercle. In dorsal view, a spiral groove is observable and continues on the medial surface of the shaft. The acetabulum is round and strongly extended (Fig. 6A). The junction between the iliac shaft and corpus ossi is slightly constricted and the ventral base of the corpus ossi possesses a preacetabular fossa. The ventral ridge of the acetabulum is high. In lateral and posterior views, the pars descendens is reduced and wide, whereas the pars ascendens is high but narrow (Figs. 6A and 6B). In ventral view, the pars descendens is broad and nearly flat. In medial view, the acetabular area is bordered by shallow ridges, between which there is, a triangular and medially prominent interiliac tubercle (Figs. 6B and 6C).

Comparison and comments

The lack of the vexillum and a poorly developed tuber superior is characteristic of the genus Bombina (Böhme, 1977). The ilium differs from Bombina orientalis by a poorly developed tuber superior. The ilium from the Selety 1A locality is distinguishable from Bombina variegata and resembles Bombina bombina in having: (1) a developed pars descendens; (2) a posteroventral ridge of the pars descendens projecting ventrally rather than posteriorly (Böhme, 1977); and (3) a developed preacetabular fossa (Sanchíz & Młynarski, 1979). We, therefore, tentatively assign the bone to Bombina bombina due to the absence of well-preserved material of the fire-bellied toads from the Selety 1A locality. The specific assignment of the ilia from the Malyi Kalkaman 2 locality is impossible due to their fragmentary preservation; therefore we describe them as Bombina sp.

The specimen from the Cherlak locality (Figs. 6D–6F) is greatly damaged with only a few observable characters remaining that allow for its identification within Bombinatoridae. The identifying characters are: (1) a large pars descendens at its anterior section, but dorsally reduced; (2) a present but larger tuber superior than that of the Maly Kalkaman 2 and Selety 1A specimens (within the family, larger tuber superior are present in the Barbatula (Folie et al., 2013)); and (3) although the ventral wall of the acetabulum is not preserved, the remaining part of its base allows for the assumption that it was markedly pronounced. Due to the incomplete preservation, the important characters needed for taxonomic identification, e.g. interiliac tubercle and junctura ilioischiadica, cannot be observed. The ilium from the Cherlak locality can, therefore, be tentatively referred to the family Bombinatoridae.

Family Pelobatidae Bonaparte, 1850

Genus Pelobates Wagler, 1830

Pelobates sp.

(Figs. 6G–6I)

Localities and material examined

Selety 1A, GIN 1110/2001-AM13, one right ilium.

Description

The corpus ossi and distal portion of the iliac shaft are present. The tips of the pars descendes and pars ascendes are broken. The bone surface is smooth and there is no tuber superior. An oblique posterolaterally–anteromedially directed spiral groove extends on the dorsal surface. Laterally, the high and long pars ascendens possesses a supraacetabular fossa (Fig. 6I). The junction between the iliac shaft and corpus ossi is not constricted. The subacetabular groove is shallow and broad. The acetabulum has a nearly triangular form, with a well-marked rim. In medial view, the corpus ilii possesses an interiliac facet with a rugose surface. The interiliac facet is composed of a large lower and a small upper portions. A well-developed interiliac tubercle is visible between these portions (Fig. 6G). The lower portion is ventroposteriorly oblique, whereas the upper one is flat, less rugose and has a concave surface. The rugose surface of the facet indicates an extensive contact between two ilia (Figs. 6G and 6H). The acetabular dorsal tuber is higher than the ventral one.

Comparison and comments

The ilium can be assigned to the family Pelobatidae based on the absence of a dorsal crest, the absence of a dorsal tubercle and the presence of an oblique spiral groove on the dorsal surface (Roček et al., 2014). The bone has the same characters of the genus Pelobates: (1) a high and long pars ascendes; (2) a well-developed spiral groove (Böhme, 2010); (3) the lack of a dorsal crest of the iliac shaft (Folie et al., 2013); and (4) a rugose surface of the interiliac facet (Rage & Hossini, 2000). However, further identification of the ilium is impossible, as it does not show relevant differences at the specific level.

Family Hylidae Rafinesque, 1815

Genus Hyla Laurenti, 1768

Hyla savignyi Audouin, 1827

Hyla gr. H. savignyi

(Figs. 6J–6O)

Localities and material examined

Novaya Stanitsa 1A, GIN 948/2001-AM20, one maxilla, GIN 948/2001-AM13, one scapula and GIN 948/2001-AM14, one sacral vertebra; Lezhanka 2A, GIN 1130/1001-AM29–AM32, four ilia, GIN 1130/1001-AM33–AM36, four scapulae and GIN 1130/1001-AM41, one trunk vertebra; Cherlak, GIN 1130/1001-AM14–AM15, two ilia; Olkhovka 1B, GIN 1111/2001-AM02, one fragmentary ilium; Pavlodar 2B, GIN 1108/2001-AM01–AM03, three ilia.

Description

The ilia from all localities resemble the same morphology, i.e. the tuber superior is dorsally and slightly laterally prominent. The tuber superior is located at the anterior corner of the acetabulum. The preserved iliac shaft is nearly cylindrical, slightly mediolaterally compressed and is devoid of crista dorsalis. The supraacetabular part of the ilium is smaller than the preacetabular. The ventroposterior margin of the iliac shaft is connected with the pars descendes by an expanded preacetabular zone, building a broad and thin lamina. The acetabulum has a nearly triangular form. The acetabular rim is prominent at its high ventroanterior edge. The posterodorsal corner of the acetabulum ascends and builds a small and prominent acetabular tuber (Fig. 6L). In medial view, the bone surface is smooth, sometimes with a shallow depression in the middle part of the corpus ossi. In distal view, the junctura ilioischiadica is slender, the acetabulum is high and the interiliac facet displays a well-pronounced ventromedial expansion. The acetabular dorsal tuber is higher than the ventral one (Fig. 6K).

The scapula, a triradiate element of the pectoral girdle, is comparatively long (Figs. 6M–6O). The bone surface is relatively smooth and is pierced by several foramina. The corpus scapulae, the middle part of the bone, is slender and long. The pars suprascapularis is preserved in a fragmentary state and most probably was not high. In dorsal view, the elongate pars acromialis is narrow and almost equal in length (Fig. 6M). The shorter and flattened processus glenoidalis is slightly broad. The processus glenoidalis and pars acromialis are separated by relatively deep sinus interglenoidalis (Fig. 6N). The margo posterior, at the corner of the processus gleinoidalis and corpus scapula, possesses an oval to elongated-oval angular fossa (Figs. 6N and 6O). The tear-shaped glenoid fossa reaches the posterior corner of the processus glenoidalis. The crista supraglenoidalis is slightly pronounced.

Comparison and comments

The Siberian fossil tree frog differs from already described fossils and some recent species of the genus Hyla. The following recent material is available for comparison: Hyla savignyi, Armenia (four individuals, unnr. GPIT specimen), Hyla orientalis, Armenia (two individuals, unnr. GPIT specimen) and Hyla arborea, Germany? (one individual, unnr. GPIT specimen). The Siberian forms can be distinguished from Hyla sp. (Rudabánya locality in Hungary, middle Late Miocene (Roček, 2005); Bois Roche Cave in France, early Late Pleistocene (Blain & Villa, 2006)), Hyla arborea (TD8 locality in Spain, early Middle Pleistocene (Blain, 2009)), Hyla cf. arborea (Mátraszőlős 2 locality in Hungary, middle Middle Miocene (Venczel, 2004)), Hyla gr. H. arborea (Capo Mannu D1 Local Fauna in Italy, Late Pliocene (Delfino, Bailon & Pitruzzella, 2011)), Hyla aff. japonica (Tologoy 38×, Baikal Lake, Russia, late Late Pleistocene (Ratnikov, 1997)) and recent Hyla japonica (Nokariya, 1983) in having: (1) a fossa supragleinoidalis; (2) a slenderer and lower corpus scapula and pars suprascapularis; and (3) a shorter and broader processus glenoidalis. Apart from these differences, the Siberian fossil tree frogs resemble Hyla sp. from the Bois Roche Cave, France (Blain & Villa, 2006), and Hyla arborea (one individual, unnr. GPIT specimen) in possessing a low and broad processus glenoidalis. The recent Hyla savignyi is the only tree frog showing a fossa supragleinoidalis like the one present in the studied remains. The recent Hyla savignyi also possesses other similarities to the fossil tree frog, namely: (1) a slender junctura ilioischiadica; (2) the same position of the tuber superior; (3) comparable acetabular tubers; and (4) a similar slightly curved pars ascendens. There are, however, also differences between the recent Hyla savignyi and the fossil tree frog. The fossil tree frog has: (1) a dorsally and slightly laterally prominent tuber superior; (2) a deeper and larger fossa supragleinoidalis; and (3) a ventromedial expansion of the interiliac facet; whereas H. savigyni has: (1) a dorsally and laterally significantly prominent tuber superior; (2) a shallow and small angular fossa; and (3) the interiliac facet devoid of ventromedial expansion. Among other fossil tree frogs, the Western Siberian Hyla sp. has the lowest and broadest processus glenoidalis. Another fossil tree frog Hyla sp. reported from the Russian Platform in the Kuznetsovka locality (0.5–0.65 Ma) (Ratnikov, 2002; Fig. 2), displays a similar morphology of the ilium as in the Siberian fossil, i.e. the orientation of the tuber superior and the form of the junctura ilioischiadica. Because of the observed differences in both the recent and fossil forms, as well as the similarities to Hyla savignyi, we assume that the fossil tree frogs from Western Siberian and the Russian Platform, probably represent a new form related to the group of Hyla savignyi.

Family Bufonidae Gray, 1825

Genus Bufo Laurenti, 1768

Bufo bufo (Linnaeus, 1758)

(Figs. 6P–6W)

Localities and material examined

Novaya Stanitsa 1A, GIN 948/2001-AM15, one left and GIN 948/2001-AM16–AM17, two right ilia, GIN 948/2001-AM18–AM19, two trunk vertebrae; Borki 1A, GIN 1115/1001-AM02, one sacral vertebra, GIN 1115/1001-AM03, one left ilium; Olkhovka 1B, GIN 1111/2001-AM04, one left, GIN 1111/2001-AM03, two right ilia and GIN 1111/2001-AM05, one trunk vertebra; Olkhovka 1C, GIN 1111/3001-AM01, one left scapula, GIN 1111/3001-AM02, one trunk vertebra and GIN 1111/3001-AM03, one urostyle; Lezhanka 2A, GIN 1130/1001-AM37, one left ilia, GIN 1130/1001-AM38, one left scapula, GIN 1130/1001-AM39, one sacral and GIN 1130/1001-AM40, one trunk vertebrae; Isakovka 1B, GIN 1131/3001-AM01, one left ilium; Isakovka 1A, GIN 1131/1001-AM01, -AM05, two right ilia; Peshniovo 3, GIN 1118/3001-AN01, one sacral vertebra; Lezhanka 1, GIN 1129/1001-AM04, one trunk vertebra; Andreievka 1, GIN 1112/2001-AM01, one right scapula.

Description and comments

The ilia are large and have a robust corpus ossi. The spiral groove is broad and very shallow. The tuber superior is broad, low, covered with irregular tubercles, and it is situated above the acetabulum (Fig. 6P). The smooth and concave pars descendens is more developed than the pars ascendens. The ventral edge of the pars descendens is thin and lamelliform. The preacetabular fossa is absent (Fig. 6P). In posterior view, the anterolateral edge of the acetabular is strongly curved. The junctura ilioischiadica shows a higher acetabular ventral tuber than the dorsal tuber, and the ventral half of the corpus ossi projects ventromedially (Fig. 6P).

The scapula is a robust bone and is longer than it is high. The material is represented by all ontogenetic series. The angular fossa is absent; a shallow groove on the ventral side of the pars acromialis is present and well pronounced in larger individuals. The pars acromialis and corpus scapula have nearly the same height. The pars suprascapularis laterally increases in height. The pars suprascapularis and corpus scapulae (anterior) have smooth surfaces. The base of the lateral edge of the fossa glenoidalis is elevated but does not project laterally. The crista supraglenoidalis is well developed in larger individuals. The anterior margin is concave. The base of the pars acromalis is high and thin (Fig. 6R). There is a shallow and expanded depression in ventral view. The anteromadial margin of the pars acromalis possesses a low tubercle. The transition from the corpus scapula to the pars acromialis is nearly straight and the wall is thin (Figs. 6S and 6T).

In several localities, the large-sized frog vertebrae and urostyle (Figs. 6U–6W) are present in association with diagnostic elements (ilia and scapula) (e.g. Olkhovka 1C locality) or are isolated (e.g. Pehsniovo 3 locality). These individuals of the same size can be assigned to the large Bufo bufo. The morphological traits described above (e.g. lack of angular fossa on the scapula and preacetabular fossa on ilium, general outline, form, and size of the scapula and ilium) as well as the bone dimensions are the same as those found in the common toad Bufo bufo (Blain, Gibert & Ferràndez-Cañadell, 2010).

Genus Bufotes Rafinesque, 1815

Bufotes viridis Laurenti, 1768

(Figs. 6X–6Z)

Localities and material examined

Baikadam, GIN 950/2001-AM02–AM04, three left and GIN 950/2001-AM05–AM09, five right ilia; Shet-Irgyz 1, GIN 1106/1001-AM01, one left ilium; Malyi Kalkaman 1, GIN 1107/1001-AM02 and -AM03, one left and one right scapulae; Malyi Kalkaman 2, GIN 1107/2001-AM03, one right scapula; Znamenka, GIN 1109/1001-AM01 and -AM02, one left and one right scapulae, GIN 1109/1001-AM03–AM07, five left and GIN 1109/1001-AM08–AM11, four right ilia; Pavlodar 1A, GIN 640/5001-AM01–AM24, 240 left and GIN 640/5001-AM25–AM58, 34 right ilia, GIN 640/5001-AM63–AM77, 15 left and GIN 640/5001-AM78–AM88, 11 right scapulae; Cherlak, GIN 1110/2001-AM16, one right ilium; Selety 1A, GIN 951/1001-AM08–AM10, three left and GIN 951/1001-AM11–AM14, four right ilia; Isakovka 1A, GIN 1131/1001-AM02–AM04, three left ilia; Kedey, GIN 951/2001-AM01 and-AM02, one left and one right ilia; Lebiazhie 1A, GIN 950/3001-AM01, one left scapula, GIN 950/3001-AM01, two left ilia; Lebiazhie 1B, GIN 950/4001-AM01, -AM02, two right ilia.

Description and comments

The iliac shaft is slightly lateromedially compressed and bears a weakly pronounced depression along the middle section. The spiral groove between the corpus ossi and iliac shaft is weakly developed. The tuber superior is low and possesses a uni- or bilabiate protuberance in its central part. The angular fossa is well pronounced. The anteroventral edge of the acetabular rim is straight. The pars descendens projects sharply ventrally. There is no observable ‘calamita’ ridge (Fig. 6X). The remains show typical features for Bufotes viridis: i.e. the form and shape of the tuber superior and acetabulum (Böhme, 1977; Blain, Gibert & Ferràndez-Cañadell, 2010). Due to the absence of well-preserved material, we prefer to tentatively assign the remains to the Bufotes viridis group.

Bufo sp.

Localities and material examined

Cherlak, GIN 1110/2001-AM17, one left scapula; Olkhovka 1A, GIN 1111/1001-AM01, -AM02, two left ilia; Pavlodar 2B, GIN 1108/2001-AM04–AM06, three left ilia.

Description and comments

The greatly damaged ilia exhibits the typical morphology of the genus Bufo, i.e. the iliac shaft without the dorsal crest and a spiral groove between the shaft and corpus ilii (Böhme, 1977). There is a preacetabular fossa in the caudoventral corner of the acetabulum. The tuber superior is eroded. In medial view, the pars descendens is ventromedially directed.

Family Ranidae Batsch, 1796

Genus Pelophylax Fitzinger, 1843

Pelophylax sp.

(Figs. 6AA–6AD)

Localities and material examined

Malyi Kalkaman 1, GIN 1107/1001-AM04, one left ilium, GIN 1107/1001-AM13, one left articular; Malyi Kalkaman 2, GIN 1107/2001-AM04, -AM05, two right ilia, and GIN 1107/2001-AM06, one left ilium; Petropavlovsk 1, GIN 952/1001-AM01, one left ilium; Olkhovka 1C, GIN 1111/3001-AM04, one right ilium; Kamyshovo, GIN 1107/1001-AM01, one right scapula; Lezhanka 1, GIN 1129/1001-AM05, one left and GIN 1129/1001-AM06, one right ilia, GIN 1129/1001-AM07, one left scapula; Andreevka 1, GIN 1112/2001-AM02, one right and GIN 1112/2001-AM03, one left ilia; Livenka, GIN 1129/2001-AM01, one right ilium.

Description and comments

The ilia have a strong, oval, nearly vertically oriented and ventrally well-defined high tuber superior. The dorsal crest is high; anteriorly it is often broken. The tuber superior is high and slightly more S-shaped than the crest; a well-developed supraacetabular fossa is present. Posterior to the tuber, the dorsal margin of the bone is bent ventrally towards the acetabulum. In posterior view, the tuber superior is curved ventromedially (Fig. 6AA). The junctura ilioschiadica is damaged, but based on the preserved structures we speculate that it was tall (Fig. 6AB).

The scapula is an elongated and short bone. In ventral view, a weakly developed crista supraglenoidalis is observable. It runs subparallel to the margo posterior and reaches the middle part of the pars suprascapulars (Figs. 6AC and 6AD).

The characters listed above, i.e. like the form and orientation of bones, tuber superior and crista supraglenoidalis, allow for the attribution of the fossils to the genus of the green (water) frogs Pelophylax (Böhme, 1977; Sanchíz, Schleich & Esteban, 1993; Bailon, 1999; Blain, Bailon & Agustí, 2007). Any further identification is impossible due to the fragmentary preservation of the material.

Genus Rana Linnaeus, 1758

Rana sp./Rana temporaria Nilsson, 1842

(Figs. 6AE–6AH)

Localities and material examined

Ayakoz, unnr. HC specimens, numerous ilia; Baikadam, GIN 950/2001-AM10, one left, GIN 950/2001-AM11–AM13, and three right ilia; Malyi Kalkaman 1, GIN 1107/1001-AM05–AM09, five left ilia, GIN 1107/1001-AM10, one right ilia, GIN 1107/1001-AM01, -AM11, two right scapula; Malyi Kalkaman 2, GIN 1107/2001-AM07, one right ilium, GIN 1107/2001-AM08–AM13, six left ilia; Olkhovka 1C, GIN 1111/3001-AM05, one right ilium; Lezhanka 1, GIN 1129/1001-AM08, one left ilium; Kentyubek, unnr. HC specimens, two left ilia.

Description

The ilia have a reduced, compact, anteriorly directed and low tuber superior. The lateral surface is rough. The dorsal crest is low. The pars descendens is more developed than the pars ascendens (Fig. 6AE). In posterior view, the junctura ilioschiadica is low (Fig. 6AF) in comparison to the ilium of Pelophylax sp. (Fig. 6AA). The tuber superior projects dorsolaterally. The pars descendens projects medially (Fig. 6AE).

The middle portions of both scapulae are preserved without the proximal parts of the pars acromialis and suprascapularis. In dorsal view, a crista supraglenoidalis is observable at the processus glenoidalis, which continues until the pars suprascapularis along the longitudinal axis of the bone. It is very prominent and forms a lamelliform convex ridge. The base of the processus glenoidalis is high and straight (Figs. 6AG and 6AH).

Comments

The ilia and scapulae morphology strongly resembles that of brown frogs genus, Rana (Böhme, 1977). Due to the fragmentary preservation of the bone material, any precise taxonomic identification of the frogs from nearly all localities was impossible. The comparison with recent species (e.g. Rana temporaria (unnr. GPIT specimen), Rana dalamtina (unnr. GPIT specimen; Bailon, 1999), Rana graeca (unnr. GPIT specimen), Rana arvalis (unnr. GPIT specimen), Rana dybowskii (MNCN 40459), Rana amurensis (unnr. GPIT specimen), etc.) revealed more similarities with the European and Western Asiatic species rather than to Eastern Asiatic brown frogs.

Only the Malyi Kalkaman 1 locality provided adequate material for specific identification. The ilia and scapulae from this locality’s material resembled the recent species, Rana temporaria, which has the widest distribution among the brown frogs in Eurasia. The fossil bones of brown frogs from other Western Siberian localities are described here as Rana sp. Due to the poor preservation of the ilia from the Kentyubek locality, it can be only identified at the family level as Ranidae indet.

Class Reptilia Laurenti, 1768

Order Squamata Oppel, 1811

Suborder Gekkota Cuvier, 1817

Family Gekkonidae Gray, 1825

Genus Alsophylax Fitzinger, 1843

Alsophylax sp.

(Fig. 7)

Figure 7 Alsophylax sp. from the localities Cherlak (A–P) and Mynsualmas-MSA 3 (Q).

(A–E) Two left dentaries; (A–D) left dentary, GIN 1110/2001-RE11; (A) mirrored labial view; (B–D) lingual view; (C) symphyseal region in lingual view; (D) the same region in ventral view, both display the symphyseal groove; (E) posterior fragment of left dentary, GIN 1110/2001-RE12, lingual view; (F–M) five maxillae; (F, G) left maxilla, GIN 1110/2001-RE26, lingual view; (H) right maxilla, GIN 1110/2001-RE39, lingual view; (I, J) right, GIN 1110/2001-RE40 and (K, L) left maxillae, GIN 1110/2001-RE27; (I, K) lingual view; (J, L) labial view; (M) left maxilla, GIN 1110/2001-RE28, labial view; (N–P) cervical vertebra, GIN 1110/2001-RE44; (N) anterior; (O) left lateral; (P) posterior views; (Q) right dentary, unnr. GPIT specimen, lingual view. Abbreviations: dl, dental lamina; ds, dental shelf; fcpr, facial process of maxilla; fMx5, foramina for mandibular division of the fifth cranial (trigeminal) nerve; hfr, haemal foramen; hl, horizontal lamella; lf, lacrimal facet; lg, longitudinal groove; lh, lamina horzontalis; mc, Meckelian canal; na, neural arch; nc, neural canal; nf, nasal facet; pfc, palatine facet; ph, paries horizontalis; prz, prezygapophysis; psz, postzygapophysis; pv, paries verticalis; pxp, premaxillary process; pyp, pterygapophysis; sac, opening of superior alveolar canal; sg, symphyseal groove; sf, splenial facet; tpr, transverse process.

Locality and material examined

Cherlak, GIN 1110/2001-RE01–RE10, 10 right dentaries, GIN 1110/2001-RE11–RE24, 14 left dentaries, GIN 1110/2001-RE26–RE38, 13 left maxillae, GIN 1110/2001-RE39–RE43, five right maxillae, GIN 1110/2001-RE44, one cervical trunk vertebra; Mynsualmas-MSA 3: one right maxilla, unnr. GPIT specimen.

Description

Tooth morphology

The teeth are slender, unicuspid, and not narrowly arranged. All maxillaries and dentary teeth are straight, except the most anterior ones on the dentary, which are anteriorly lightly oblique. The central teeth on dental lamina of both the maxilla and dentary are larger than the anterior and posterior ones (Figs. 7C and 7G). The cusps of maxilla teeth are rarely posteriorly oriented. The most complete dentary bone contains at least 17 (in total, probably 20) teeth, counted by both teeth and their alveoles (Figs. 7B–7D).

Dentaries

The dentary is a slender and elongated. In the symphyseal region, the bone is slightly medially curved. The pars ventralis is assumed to be enlarged, due to the bone posteriorly increasing in height. The dentary is characterised by a completely closed Meckelian canal, which runs along approximately two-thirds of the bone length (Fig. 7B). The symphyseal articulation surface is reduced. It does not build a pronounced articulation surface. The ventral surface of the symphysis bears a longitudinal, posteriorly deepening symphyseal groove, visible in both the lingual and ventral views (Figs. 7B–7D). The Meckelian canal is open posteriorly at about the 15th–16th tooth position. The splenial facet on the dentary, the anterior margin of Meckelian opening, shows a light concave and elongated surface (Figs. 7B–7E). In lateral view, the bone is smooth, and the only complete dentary possesses five foramina that are arranged in a longitudinal row (Fig. 7A). The size of the foramina increases slightly in the anteroposterior direction, also changing in form from a more rounded outline to an oval appearance. The position of the last mental foramen is arranged lingually in front of the posterior opening of the Meckelian canal. The cavity of the Meckelian canal is divided in two, i.e. the upper and lower subcanals, by a distinct horizontal lamella (Fig. 7E). The horizontal lamella runs parallel to the lamina horizontalis and can be observed posteriorly behind the opening of the Meckelian canal. The upper subcanal opens to the labial surface of the bone near to the mental foramina. The symphyseal groove corresponds to the anterior opening of the lower subcanal. In lingual view, the lamina horizontalis is situated in a low position. Its margin is rounded but not prominent. A shallow and anteriorly extending dental shelf divides the lamina horizontalis from the dental lamina (Fig. 7C). Posteriorly, the bone is nearly L-shaped in the transverse section. In all observed specimens, the pars horizontalis is destroyed in the preserved bone. The caudal portion of the paries verticalis shows bifurcation (Fig. 7E), which corresponds to the coronoid insertion.

Maxilla

The preserved posterior part of the maxillary possesses a relatively low lacrimal facet of the facial process of the maxilla (pars nasalis sensu Estes (1969)), while the latter is always not preserved. The internal wall of the maxilla posteriorly bears a small distinct longitudinal groove, running parallel to the lamina horizontalis (Figs. 7F–7H). The groove begins at the posterior basis of the lacrimal facet and continuous until the preserved posterior tip of the bone. The groove narrows at the middle section of the bone (at the position of the third to fourth last tooth), where the lacrimal facet terminates. The lamina horizontalis is clearly visible, expands laterally just under the tip of the lacrimal facet and builds a palatine facet (Figs. 7F–7H). The lamina horizontalis becomes distinctly and posteriorly narrower but does not diminish fully at the posterior end of the bone. The jugal process of the maxilla is bifurcated at its distal end (Fig. 7H). The maxillary lappet is damaged, but its base is preserved. The internal wall surface of the maxilla contains few rugosities. Here, an anteroposteriorly directed, fairly well-pronounced, median ridge, is observed. In labial view, several foramina occur above the dental row. Some of these foramina are arranged in a longitudinal line that corresponds to the foramina for the mandibular division of the fifth cranial (trigeminal) nerve. This line runs parallel to the lamina horizontalis. The last foramen in the row pierces the maxilla at the base of the lacrimal facet under its tip. The bases of the facial process and maxillary lappet lay a relatively large superior alveolar canal (sac, Figs. 7J and 7L) for the maxillary nerve and its accompanying blood vessel. The remaining foramina at the maxilla are dispersed irregularly on the bone surface. The premaxillary process is present, but it is highly damaged. The anterior basis of the lacrimal facet is pierced by a foramen.

Vertebra

A single cervical vertebra of a gecko specimen shows an elongate amphicoelous centrum (Figs. 7N–7P). The cotyles are approximately circular. In anterior view, the vertebra has a semi-circular outline. In lateral view, the vertebra is anteroposteriorly compressed. The neural arch is concave on both sides. The transverse processes are high, extremely short, and vertically aligned. The distal end of the process is round. The haemal foramina are present at the lower base of the transverse processes. The prezygapophyses are small and slightly prominent. The neural arch is plane and triangular in outline. It possesses a slender and low neural crest. The postzygapophyses are small, nearly invisible, and are situated on the ventrolateral edges of the pterygapophysis.

Comparison and comments

Numerous characters allow for the identification of the material as a member of the family Gekkonidae. These characters are namely: (1) the amphicoelous condition of the vertebra; (2) the maxillae and dentaries bearing numerous pleurodont, isodont, densely packed, cylindrical, and slender monocuspid teeth; (3) the presence of a medially extended dental shelf of the maxilla; and (4) the lingually closed Meckelian canal (Hoffstetter & Gasc, 1969; Daza, Alifanov & Bauer, 2012). The gekkonid remains from the Cherlak locality display a low number of teeth on the dentary (up to 20) and a rounded tooth apex (making the teeth digitiform), which are diagnostic characters for the genus Alsophylax (Nikitina & Ananjeva, 2009). Within the gekkonids, a low number of teeth (up to 20) is also characteristic of Mediodactylus russowi, Phelsuma laticauda, and Phelsuma serraticauda (Nikitina, 2009). The Siberian fossil geckos can be distinguished from Mediodactylus by peculiarities of the maxilla (i.e. the presence of a lingual longitudinal groove and a reduced row of foramina of the trigeminal nerve), the dentary with a distinct and longer horizontal lamella, plus a reduced symphyseal groove. The recent genus Phelsuma can be excluded from consideration since these geckos are restricted to the islands of the southwest part of the Indian Ocean and belong to another zoogeographic zone. The fossil geckos resemble the recent species Alsophylax pipiens (see in Estes (1969); Table 2C) in the presence of the prefrontal process and their short row of foramina of the trigeminal nerve, which terminates below the prefrontal process. Further comparison with the recent genus Alsophylax is, however, impossible due to the lack of available comparative osteological material of the recent Alsophylax species.

Fossil geckos were present in the Early Miocene of Kazakhstan, as is evident from the Mynsualmas-MSA 3 locality (unnr. GPIT specimen) (Böhme & Ilg, 2003). The re-studying of the material revealed that the posterior fragment of a right maxilla shows morphology similar to Alsophylax sp. from the Cherlak locality in the presence of a lingual longitudinal groove, the absence of foramina at the posterior portion of the bone and a round tooth apex. The fossil material, however, differs in its larger size (Fig. 7Q). Taking this difference as well as the similarities into account, we tentatively consider the Mynsualmas record as cf. Alsophylax sp. This fossil probably represents a larger Alsophylax species than those registered in the Western Siberia.

Suborder Lacertilia Owen, 1842 sensu Estes, Queiroz & Gauthier, 1988

Family Lacertidae Fitzinger, 1826

Genus Lacerta Linnaeus, 1758

Remarks

The generic assignment of fossil lacertid remains is extremely difficult. This group is anatomically generalised (Lacera sensu lato) and shows very few characteristic features (e.g. bone and teeth morphology) for detailed taxonomic assignments (Böhme, 2010; Böhme & Vasilyan, 2014).

Lacerta s.l. sp. 1.

(Fig. 8A)

Figure 8 Lizard and turtle remains from the Western Siberian localities.

(A) Lacerta s.l. sp. 1, left dentary, Pavlodar 1A, GIN 640/5001-RE01, lingual view; (B) Lacerta s.l. sp. 2, right dentary, Pavlodar 1A, GIN 640/5001-RE34, lingual view; (C, D) Eremias sp., frontal, Pavlodar 2B, GIN 1108/2001-RE01; (C) dorsal view; (D) ventral views; (E) Emydoidea sp., fragment of right hypoplastron, GIN 948/2001-RE01, ventral view; (F, G) Emydoidea sp., left femur, GIN 948/2001-RE02; (F) cranial view; (G) ventral view. Scale bars: A, C, D = 2 mm; B = 1 mm; E–G = 1 cm.

Material

Baikadam, GIN 650/2001-RE07–RE09, two (3?) left dentaries, GIN 650/2001-RE10, one postsacral vertebra; Pavlodar 1A, GIN 640/5001-RE01–RE15, -RE41–RE4217 left dentaries, GIN 640/5001-RE16–RE25, 10 right dentaries.

Description

The bones bear pleurodont bicuspid teeth. The most completely preserved dentary possesses at least 20 teeth. The pars dentalis is tall, with its height corresponding to two-thirds of the tooth length. The Meckelian groove is open ventrolingually. It starts from the ventral side of the symphysis and posteriorly increases in height. The lamina horizontalis is slightly curved. The anterior portion of the lamina horizontalis is high and broad, reaching its maximal height in its middle section, which corresponds to the 10th tooth position. Behind this point, the lamina horizontalis articulates ventrally with the dorsal margin of the splenial and gradually narrows posteriorly. The articulation surface is lingually exposed. The crista dentalis, sensu Roček (1984), is not higher but is longer than the ventral margin of the lamina horizontalis. The ventral margin of the crista dentalis, in its posterior half, bears an articulation surface with the ventral margin of the coronoid. A lingually exposed articulation surface of the splenial is located at the posterior portion of the ventral surface of the lamina horizontalis. Up to eight small foramina are present in labial view (Fig. 8A).

Comments

See in Lacerta s.l. sp. 2.

Lacerta s.l. sp. 2.

(Fig. 8B)

Material

Pavlodar 1A, GIN 640/5001-RE27–RE33, seven left dentaries, GIN 640/5001-RE34–RE39, six right dentaries; Cherlak, GIN 1110/2001-RE51, one right dentary.

Description

The dentaries possess 19 bicuspid teeth. The pars dentalis is high with its height corresponding to two-thirds of the teeth length. The lamina horizontalis is curved and maintains almost the same height along its entire length. The lamina horizontalis decreases slightly in height only at the 9th–10th tooth positions, where the splenial articulates with the lamina horizontalis. The articulation facet is lingually exposed only in its most posterior portion. The crista dentalis is short but is longer than the ventral margin of the lamina horizontalis. The Meckelian groove is low and ventrolingually open. Up to seven small foramina are present in labial view (Fig. 8B).

Comments

Lacerta s.l. sp. 2 differs from Lacerta s.l. sp. 1 in having: (1) a more curved lamina horizontalis that maintains nearly the same height along its length; (2) a higher and broader anterior portion of the lamina horizontalis; (3) a shorter cirsta dentalis; and (4) a lower Meckelian groove.

Lacerta s.l. sp./Lacertidae indet.

Material

Malyi Kalkaman 2, GIN 1107/2001-RE01, one vertebra; Olkhovka 1A, GIN 1111/1001-RE01 and–RE02, one anterior and one posterior trunk vertebrae; Cherlak, GIN 1110/2001-RE06, -RE52–RE57, seven trunk vertebrae, GIN 1110/2001-RE47, -RE48, two left maxillae, GIN 1110/2001-RE49, one right maxilla, GIN 1110/2001-RE50, one left dentary; Pavlodar 1A, GIN 640/5001-RE40, one premaxilla, GIN 640/5001-RE26, numerous fragments of dentaries and maxillae, GIN 640/5001-RE43, 77 vertebrae; Pavlodar 1B, GIN 640/6001-RE01, -RE02, two left dentaries, GIN 640/6001-RE03, -RE04, two right dentaries; Olkhovka 1B, GIN 1111/2001-RE01, one right dentary; Pavlodar 3A, GIN 1108/3001-RE01, one right maxilla; Beteke 2, GIN 945/6001-RE01, one left dentary; Beteke 4, GIN 945/8001-RE01, one left dentary.

Description and comments

The preserved maxillaries and dentaries possess pleurodont bicuspid teeth. The Meckelian groove is lingually open. The labial surfaces of the maxillaries show no ornamentation. In labial view, the foramina for mandibular division of the fifth cranial (trigeminal) nerve are observable. They are situated along a longitudinal line, parallel to the ventral margin of the bone. The opening of the superior alveolar canal is large. In lingual view, a shallow but broad groove is present at the anterior portion of the frontal process. The large foramen of the fifth cranial (trigeminal) nerve opens at the ventral surface of the lamina horizontalis. A single premaxilla from Pavlodar 1A, GIN 640/5001-RE40 has a tapering nasal process with a row of seven pleurodont and monocuspid teeth.

The bone material is extremely fragmentary, and any comparison between different localities was impossible. The fossil remains (maxillae and premaxilla) from Pavlodar 1A do not show any taxonomical differences, so we were not able to group them neither to Lacerta s.l. sp. 1 nor Lacerta s.l. sp. 2. Besides the jaw material, vertebrae from the trunk region are available in the Maly Kalkaman 2, Olkhovka 1A, and Cherlak localities. It was not possible to identify all of remains below the family level.

Genus Eremias Fitzinger, 1843

Eremias sp.

(Figs. 8C–8D)

Material

Pavlodar 2B, GIN 1108/2001-RE01, -RE02, one frontal and one trunk vertebra.

Description

The preserved frontal has a sandglass shape and the most anterior and posterior portions are broken. The bone is slightly curved in lateral view. The posterior portion of the dorsal surface is rough. The crista cranii are round and slightly elevated at the narrowest portion of the bone. Anteriorly, these crista cranii increase in height and build the lateral walls of the cranial vault. The anteroventral surface of the bone has two drop-shaped grooves. The posteroventral surface is plain and slightly lower than the anterocentral surface. The prefrontal facets are developed but do not show any lateral extension. The bone margin that connects both facets is concave. In dorsal view, the nasal facets that are situated at the anterolateral corners, are narrow, deep, and elongated (Figs. 8A and 8D).

In lateral view, a single preserved trunk vertebra has a rectangular shape. The neural arch is moderately convex. A narrow and deep groove is present at the transition of the neural arch and prezygapophysis. The neural spine is reduced and posteriorly builds a rounded process, projecting over the posterior margin of the arch. The centrum is compressed anteroposteriorly and possesses two shallow subcentral grooves, with a subcentral foramina in each one. The condyle is small, round and is situated in the middle part of the posterior margin of the centrum.

Comments

Among the Eurasian lacertids, fused dorsally sculptured frontals are known in Acanthodactylus, Eremias, Ophisops (Evans, 2008). Our own observations of recent species of these genera (Eremias strauchi, Eremias pleskei, Eremias arguta, Eremias multicellata, Ophisops elegans, Acanthodactylus erythrurus) allowed for the assignment of the frontals to the genus Eremias and to separate them from: (1) Ophisops by a robust frontal, more pronounced grooves at the anteroventral bone surface and a lack of the lateral extension of the prefrontal facet; and (2) Acanthodactylus by a flat posteroventral surface of the bone and a less curved outline in lateral view. The preserved single vertebra strongly resembles the morphology that is found in Eremias (Rage, 1976).

Order Testudines Linnaeus, 1758

Suborder Cryptodira Cope, 1868

Family Emydidae (Rafinesque, 1815)

Genus Eymdoidea Gray, 18702

Emydoidea sp.

(Figs. 8E–8G)

Material

Novaya Stanitsa 1A, GIN 948/2001-RE01, one posteriorly incomplete right hypoplastron, GIN 948/2001-RE02, one left femur.

Description and comments

The caudal part of the left hypoplastron, which has a width of 54.3 mm, is preserved (Fig. 8E) and probably belongs to a middle-sized individual with a total length of the carapace, approximately 300 mm. In ventral view, the femoral/abdominal sulcus is nearly straight and curves anteriorly only near the lateral edge of the bone, terminating at the base of the inguinal buttress. The bone is comparatively thin, medially from the bridge (4 mm) to behind the bridge (7.2 mm). The lateral edge of the bone projects slightly posterolaterally. The outline of the femoral/abdominal sulcus and the profile of the lateral edge are similar to those of the emydid genus Emydoidea (both fossil and recent specimens) (Chkhikvadze, 1983; p. 138, Figs. 26 and 27; Holman, 1995).

An almost complete left femur is available from the same locality where the hypoplastron fragment was found. The bone is slender and bent (Figs. 8F and 8G), and is 50.6 mm in length. This bone could have belonged to an individual of about 300 mm of the carapace length. The femur lacks its proximal portion (i.e. femoral head, major and minor trochanters). In ventral view, the fossa is delimited by the trochanters and is observable below the femoral head. The dimension of the bone is characteristic of aquatic testudinoids. Taking this latter character into account, as well as the comparable reconstructed total body-sizes of both elements (hypoplastron and femur) (ca. 300 mm), we consider the remains to belong to the genus Emydoidea.

Testudines indet.

Material

Malyi Kalkaman2, GIN 1107/2001-RE02, shell fragment; Shet-Irgyz 1, GIN 1106/1001-RE01, one neuralia; Petropavlovsk 1, GIN 952/1001-RE01, several fragments of carapax; Borki 1B, GIN 1115/2001-RE01, one fragment of carapax.

Comments

The preserved remains were not sufficiently informative for any other taxonomic interpretation.

Discussion

Neogene evolution of amphibian and reptile assemblages in Western Siberia

In general, and in contrast with the well-studied European fossil record, very little is known about the Neogene herpetofauna from Asia. This record bias is owing to: (1) the less explored and less extensively studied Neogene deposits on the Asian continent; and (2) the entirely lack of study on recent amphibians and reptiles, in spite of the intense investigations around small mammals by many scholars. The Western Siberian localities provide an exceptional opportunity to fill these gaps in information and to explore both the unknown diversity of the Asian herpetofaunal assemblages and the palaeobiogeographic affinities of the Western Siberian Neogene herpetofauna with the European faunas. Unfortunately, the yielded fossil material from this study and from previous investigations has thus far not been rich in amphibian and reptile remains. On average, only four taxa are available from each studied locality. Our faunistic, palaeogeographic and palaeoclimatic interpretations are, hence, very tentative and should be taken within this context. The unbiased comparison and analysis of our data are also hindered by the scarce record of the Asian Neogene fossil fauna. For the comparison with the European record, we used already published data on amphibian and reptile groups (families, genus, species, etc.) which have been summarised in the fosFARbase database (Böhme & Ilg, 2003). These data are given in Table S5. In the ‘Europe’ record, we consider all known fossil records from Western, Central, and Eastern Europe as well as from Anatolia (Fig. 9). By analysing the Neogene amphibian and reptile records from Europe and Asia, we were able to provide useful data that are applicable for fossil calibration of molecular clocks in the phylogenetic trees.

Figure 9 The European (Supplemental Information 3) and Western Siberian (present study) Neogene fossil record of the studied amphibian groups.

Detailed list of the localities see Supplemental Information 3 and for the family Cryptobranchidae—Böhme, Vasilyan & Winklhofer (2012; Table 1). The occurrences of each group in Europe and Western Siberia are given in the same colour. The Paleogene records of the groups are indicated by arrows. Abbreviations: Hyn, Hynobiidae; Cry, Cryptobranchidae; Prot, Proteidae; Chel, Chelotriton; Tylt, Tylototriton; Bomb, Bombina; red balk, Bombina (cf.) variegata; black balk, Bombina (cf.) bombina; Palbr, Palaeobatrachidae; Pelb, Pelobatidae; Hyla, Hyla; white balk, Hyla (cf.) arborea; Bbuf, Bufo bufo (group); Bvir, Bufotes (cf.) viridis/group of Bufotes viridis; Rana, Rana (cf.) temporaria; Pelx, Pelophylax.

Hynobiidae

The Asiatic salamanders (Salamandrella sp.) have the most abundant and frequent record among the studied Western Siberian localities. These organisms appeared in these areas in the late Middle Miocene (in the Malyi Kalkaman 1 locality) and are present until the middle Early Pleistocene. Although the herpetofaunal assemblages of the older localities are rich and represented by numerous taxa, they do not contain any hynobiid remains, demonstrating that there is no sampling bias in their record and that such specimens are not present in earlier localities.

Recently, the oldest record of the genus, Salamandrella sp. has been described from the late (?) Early Miocene of Eastern Siberia (Lake Baikal) (Syromyatnikova, 2014), and a new species of Salamandrella is indicated to be present in the Late Miocene locality Ertemte 2, China (Vasilyan et al., 2013). Furthermore, the fossil Asiatic salamander, Ranodon cf. sibiricus was recovered from the Early Pleistocene of Southern Kazakhstan (Averianov & Tjutkova, 1995), and a Salamandrella sp. was reported from a few Middle Pleistocene age localities in European Russia (Ratnikov, 2010).

In Central Europe, hynobiids (genus Parahynobius) appeared at the earliest Late Miocene and is present in the record until the Middle Pleistocene (Venczel, 1999a, 1999b; Venczel & Hír, 2013). According to the personal observations of one of the co-authors of this study (Davit Vasilyan, 2015), the hynobiids are also present in three Ukrainian localities—Grytsiv (11.1 Ma), earliest Late Miocene; Cherevichnoe lower level, middle Late Miocene; and Kotlovina lower level, Late Pliocene. The Ukrainian occurrences coincide with both the Central European and Western Siberian records of hynobiids, which at that time most probably characterised by favourable conditions for hynobiid distribution. Considering their oldest records, the origin of Hynobiidae was most probably in Eastern Asia in the Early Miocene. We will present a detailed study on the Cenozoic record of fossil Hynobiidae including the Western Siberian material in a forthcoming paper.

Cryptobranchidae

The cryptobranchid remains are known from two localities in the town of Pavlodar and from three localities in the Zaisan Basin. The stratigraphic positions of the Pavlodar localities are not clear. The only record of giant salamander that we were able to study is stored at the Palaeontological Institute of Moscow, Russia. The collection label provides the following information: ‘collected by Gaiduchenko, in 1970, from the Gusiniy Perelet locality, at the contact of the Aral clays with overlying sands, about 200–300 m south far from the “Gusiniy Perelet” [=Pavlodar 1A] locality’. The only explanation of the stratigraphic allocation of the giant salamander remains is that they originated from the basal horizon of the Pavlodar Svita, overlaying the ‘Aral clays’ (or = limnic clays of the Kalkaman Svita). Gaiduchenko (1984) and Gaiduchenko & Chkhikvadze (1985) mention a giant salamander (Cryptobranchidae indet.) from a locality named Detskaya Zheleznaya Doroga (engl. Children Railway) (Fig. 2; Table S1; Data S3), a sand pit located 10 km southeast from the ‘Gusiniy Perelet’ [=Pavlodar 1A] locality. The age of this fossiliferous horizon may fall near the Miocene–Pliocene boundary, an assumption that is mostly based on geology, age and accompanying fauna (see Data S2). This record from the Detskaya Zheleznaya Doroga presents the most northern (52.3° N) occurrence of the giant salamanders in the Northern Hemisphere known so far. Unfortunately, this material was not available for our study.

Giant salamander remains have also been reported from three Burdigalian localities—Tri Bogatyrya, Vympel, Poltinik of the Zaisan Basin (Fig. 1; Table S1) (Chkhikvadze, 1984; Böhme, Vasilyan & Winklhofer, 2012). The remains were assigned to the species Andrias karelcapeki by Chkhikvadze (1984). The taxonomic validity of the species still requires revision, which is necessary for any further interpretations.

Proteidae

The oldest record of the genus is described as being from the late Oligocene and was found in the Aral Formation in the Akespe locality, on the north coast of the Aral Sea, Kazakhstan (cf. Mioproteus,) (Malakhov, 2003; Bendukidze, Bruijn & Van den Hoek Ostende, 2009). Here, we add to the record a new, more recent Miocene (Aquitanian) Asian occurrence from the Ayakoz locality, Kazakhstan (Fig. 3D; Table S1). In the Middle Miocene, representatives of this genus occur in several localities in Southern Russia and Northern Kazakhstan (Table S1). According to our assessment, proteids survived until latest Miocene/earliest Pliocene (locality Petropavlovsk 1/2). The oldest stratigraphic record of Mioproteus (Mioproteus caucasicus) in Europe is described from the mid Aquitanian (early Early Miocene about 20.5–22 Ma) at two localities Ulm-Uniklinik and Ulm-Westtangente of the North Alpine Foreland Basin (Heizmann et al., 1989). The fossil proteids are known in Europe until the Pleistocene Epoch (Böhme & Ilg, 2003). Due to the lack of complete fossil skeletons and unclear taxonomic assignments of the fossil records, Malakhov (2003) preferred to refer all known specimens of Mioproteus to the ‘Mioproteus caucasicus complex’, including Mioproteus from Ashut, Kazakhstan, Mioproteus caucasicus from type locality, as well as from the Late Miocene of Czech Republic, Mioproteus wezei from the Pliocene of Poland and from the Early Pleistocene of Moldavia (Malakhov, 2003). Later, Roček (2005) considered Mioproteus wezei as a junior synonym of Mioproteus caucasicus, although as already mentioned by Malakhov (2003), an adequate amount of material including cranial and postcranial elements is necessary to solve the taxonomic problems of the genus. Malakhov (2003) also suggested an Asiatic origin for the ‘Mioproteus caucasicus complex’ and their later distribution into Europe. In summary, the oldest Late Oligocene record of Mioproteus (Mioproteus sp.) from Akespe, Kazakhstan and other localities of younger ages suggest: (1) a probable Asian origin of the genus; (2) the genus was continuously present in Central Asia/Western Siberia until the Early Pliocene; and (3) Mioproteus migrated into Europe in the Early Miocene.

Salamandridae

As has already been established, Chelotriton is a basket taxon (Böhme, 2008) that needs further taxonomic study. It is one of the fossil amphibians that has an abundant and wide distribution in the late Paleogene and Neogene localities of Europe. In Asia, the genus was previously known only from the late Middle Miocene locality Malyi Kalkaman 1 (Tleuberdina et al., 1993). Our study showed that this genus was present at least since the Aquitanian age (the Aykoz locality in Kazakhstan, Early Miocene) (Table S1), making their Asiatic record older than previously assumed.

Two localities (Ayakoz and Baikadam) from Western Siberia revealed aff. Tylototriton. The vertebrae showed significant similarities with the recent East Asiatic genus Tylototriton. In Böhme & Ilg (2003) and Böhme (2010), the genus Tylototriton (cf. Tylototriton sp. and Tylototriton sp. nov.) has been reported from several Early Oligocene localities in Southern Germany. Two Siberian records represent the first fossil occurrence of the genus in Asia, which appeared more recently in the fossil record than in the European occurrence. These Western Siberian specimens and the European specimens can be clearly separated from each other by the morphology of the trunk vertebrae. The Siberian salamanders probably represent new forms, strongly related to the East Asian terrestrial salamander, Tylototriton. The sympatric occurrence of two fossil terrestrial salamander genera Chelotriton and Tylototriton was documented for the first time from the Aquitanian age locality Ayakoz.

Palaeobatrachidae

The palaeobatrachids are considered a European family, with probable occurrence in North America at the terminal Cretaceous (Wuttke et al., 2012). Records of the palaeobatrachids are known from the Paleogene of Western and Central Europe. It should be taken into account, however, that records from the Paleogene of Turkey, as well as from the Paleogene and Early to Middle Miocene of Eastern Europe, are very scarcely known. In the Miocene, palaeobatrachids appear to have expanded their distribution to Eastern Europe and also reached Anatolia, where they existed from the latest Oligocene and remained during the entire Early Miocene. During the Middle Miocene, palaeobatrachids were present in Europe, from Germany to Ukraine (Wuttke et al., 2012). The palaeobatrachid record in Europe is characterised by a four-million-year-long (ca. 5.6–9.78 Ma) gap in the Late Miocene (Fig. 9). During this gap, no palaeobatrachid is known from Western to Eastern Europe even in localities rich in diverse herpteofaunal assemblages (e.g. Staniantsi, Bulgaria; Morskaya 2, Russia, Böhme & Ilg, 2003) and where characterised by favourable environmental conditions for their distribution. After this gap, palaeobatrachids occur near the Mio–Pliocene transition in studied localities from Italy (Ciabot Cagna) (Cavallo et al., 1993) and Hungary (Osztramos 1C) (Venczel, 2001)). They seem to have disappeared from Western (Tegelen locality in Holland, Villa et al., 2016) and Central Europe (Betfia IX/B locality in Romania, Venczel, 2000) after the Early Pleistocene and remained exclusively in Eastern Europe until the middle Pleistocene (Poland–European Russia) (Wuttke et al., 2012). The palaeobatrachids appear to have never reached the east of the Ural Mountains. Their most eastern distribution is recorded in the Late Pleistocene locality of Apastovo, in Russia, which is about 600 km west from the Ural Mountains (Wuttke et al., 2012). The Western Siberian record does not only represent the first and only out-of-Europe occurrence of the family, but, surprisingly, falls within the Late Miocene palaeobatrachid gap of the European record. It is possible that palaeobatrachids occupied Western Eurasia again at the Mio–Pliocene boundary, from the east.

Bombinatoridae

The primitive family of aquatic toads Bombinatoridae includes two recent genera: Bombina and Barbourula. The family is known since the Maastrichtian, Late Cretaceous in Romania, genus Hatzegobatrachus (Venczel et al., 2016) and the early Eocene in India, genus Eobarbourula (Folie et al., 2013). The recent distribution of Bombina is confined to continental Europe and East Asia, representing the western and eastern genetic clades of the genus respectively. In Europe, two species Bombina bombina and Bombina variegata are known. Bombina bombina has the widest distribution and is found in Central to Eastern Europe, whereas Bombina variegata occurs in Central Europe and in the southeastern and western parts of Eastern Europe (Pabijan et al., 2013). The fossil record of the fire-bellied toad Bombina is patchy and limited to the Neogene of continental Europe. According to Sanchiz & Schleich (1986), the oldest fossil occurrences of the genus (Bombina sp.) are known from two localities in Germany: Weißenburg 6 (Early Aquitanian) and Stubersheim 3 (Early Burdigalian) (Sanchiz & Schleich, 1986; Böhme & Ilg, 2003). The personal observations of one of the co-authors of this study (Madelaine Böhme) did not confirm the Weißenburg 6 record of Bombina. Therefore, in the present study, we consider Stubersheim 3 to be the earliest occurrence of the genus.

Bombinatorids later appeared in Central Europe in the mid Middle Miocene (Bombina sp., Opole 2, Poland) (Młynarski et al., 1982). Later, fire-bellied toads are present in three localities, representing the middle Tortonian age, including also the first fossil occurrences of the recent European species—Bombina sp. from Rudabánya in Hungary (9.9–10.30 Ma) (Roček, 2005), B. cf. bombina from Kohfidisch in Austria (8.55–8.95 Ma) (Tempfer, 2005), and Bombina cf. variegata from Suchomasty in Czech Republic (8.8–9.2 Ma) (Hodrová, 1987). During the Pliocene, bombinatorids are represented mainly by the species Bombina bombina in six localities within Central Europe (Böhme & Ilg, 2003). The Pleistocene record is the richest in bombinatorid specimens with record from over 15 localities ranging from Central to Eastern Europe, and in which both recent European species, Bombina variegata and Bombina bombina, are documented (Böhme & Ilg, 2003) (Fig. 9; Table S5).

In Western Siberia, bombinatorids are known from three localities: Malyi Kalkaman 2, Selety 1A and Cherlak. The oldest known record dated back to the late Serravalian (Middle Miocene). The oldest Messinian Selety 1A locality provided the fossil form of the recent Bombina bombina (Bombina cf. bombina) (Fig. 9). The last record of the genus dates back to the early Messinian (Late Miocene). It is interesting to note that the Western Siberian record of the genus does not coincide with their European occurrences, i.e. they are present during those periods when Bombina is absent in Europe. According to our analysis, it is clear that the ancestor of the ‘Bombina bombina–Bombina variegata’ clade was present in Europe from, at least, the later part of the Early Miocene. Later in the Middle Miocene, they expanded into Western Asia, reaching the east from the Ural Mountains. The Western Siberian fossil Bombina can be clearly osteologically separated from Bombina orientalis, a member of the East Asian clade of the genus. Taking their recent distribution as well as the fossil record into account, a split of the European and Asian Bombina clades seems most probable in Asia during the Paleogene.

Pelobatidae

The family of European spadefoot toads Pelobatidae includes only one extant genus with four species distributed in Northwestern Africa, Europe, in small areas that are east of the Ural Mountains in Russia and in the north of Kazakhstan (Kuzmin, 1995). The family has Laurasian affinities and records are known from the Late Cretaceous in North America. The presence of pelobatids in Europe dates back to the early Eocene, as indicated by the fossil genus Eopelobates (middle Eocene–Late Pliocene), as well as by the fossil forms of the recent genus Pelobates (middle Oligocene–recent) (Roček et al., 2014). The Asian record of Pelobatidae is very limited and includes forms from the Eo-Oligocene of Kazakhstan (Chkhikvadze, 1985) and Eocene of India (Folie et al., 2013). Recently, Roček et al. (2014) excluded the genus Uldzinia (Oligocene, Mongolia) (Gubin, 1995) from the family Pelobatidae. The Kazakhstan fossil record of the family (Chkhikvadze, 1985, 1998) includes numerous remains of Pelobatidae indet. from: (1) the localities of the Zaisan Basin from the Upper Aksyr Svita,3 early Priabonian; rare finds in the Kusto Svita and basal horizon of Buran Svita,4 late Priabonian and earliest Rupelian; abundant occurrences in the Buran Svita,5 early Rupelian and (2) large-sized spadefoot toads from the Kyzylkak locality of the Turgay Basin, Central Kazakhstan, late Oligocene (Chkhikvadze, 1998). Revision of this rich pelobatid record from the Zaisan Basin was not possible due to the lack of descriptions and illustrations of the material as well as the difficulty in accessing the specimens. Nevertheless, taking the Paleogene fossil records into account, we inferred that the spadefoot toads may have dispersed from Europe to Western Asia during the late Eocene to early Oligocene. It cannot be ascertained if the Pelobates sp. from the Selety 1A (early Messinian, Miocene) is a European or Asian migrant.

Hylidae

The family of tree frogs, Hylidae, has a wide distribution in Eurasia and is represented by the monophyletic genus Hyla. The most recent phylogenetic study of the genus Hyla by Li et al. (2015) recognised two closely related clades in Eurasia, namely the West Palaearctic arborea-group and East Palaearctic chinensis-group, as well as a small East Palaearctic japonica-group that is related to the North American clade of Hyla. The revision of the Western Eurasian Hyla phylogeny, based on molecular genetic studies, revealed a high diversity in the area containing about eight (?nine) (Li et al., 2015) or 10 (Gvoždík et al., 2010) species. Among them, there are two clades: (1) Hyla savignyi in the east (Levant and the area of Turkey, Iran, Armenia, Georgia) and (2) Hyla arborea (Western, Central Europe, and Balkan) + Hyla orientalis (southeastern Europe, Georgia, Armenia, Iran), which have wide distributions in the east and west respectively (Stöck et al., 2008a; Gvoždík et al., 2010).

The oldest European record of the genus is known from the Oberdorf O4 locality, late Early Miocene, Austria (Sanchíz, 1998b). After an interruption/gap of approximately three million years, records of the genus continued in the late Langhian with the first fossil appearance of the recent species Hyla arborea (Hyla cf. arborea, Mátraszőlős 2, Hungary) (Venczel, 2004). The record is almost consistent in the entire Neogene and Quaternary periods of Europe (Fig. 9). There is quite an abundant record of the genus with the oldest and first occurrences of Hyla savignyi (Hyla cf. savignyi) derived from five localities in Western Siberia, dating back to the late Late Miocene and early Early Pliocene. Apart from the distribution in Siberia, Hyla savignyi also may occur in Southern Russia, in the Middle Pleistocene (Ratnikov, 2002) (see ‘Comparison and Discussion’ in Hyla gr. Hyla savignyi), representing the youngest fossil record of the species.

Based on the fossil record of the tree frogs, we concluded that two large Western Eurasian clades split in Europe during the Middle Miocene. Our data indicated older ages for the first fossil occurrences of these clades than has been previously estimated from molecular data in two recent studies (Gvoždík et al., 2010; Li et al., 2015). Gvoždík et al. (2010)6 suggested that the split of Hyla orientalis/arborea and Hyla savignyi occurred 11.1 Ma (early Late Miocene, early Tortonian), which is approximately three million years younger than the first fossil occurrence of Hyla cf. arborea (Table S5). Whereas, without calibrating the molecular clock using the oldest European fossil Hyla (Hyla sp., Oberdorf O4 locality in Austria), Li et al. (2015) estimated this split to have occurred at 12–20 Ma, during a time interval in which the oldest fossil tree frogs related to the recent Hyla arborea occurred. In both of the cases, the interpretation of the molecular phylogeny of the group can be improved by calibrating the phylogenetic tree with the fossil record introduced in this study.

Considering our data and the results presented by Li et al. (2015), we suggest the following distribution pattern for the West Palaearctic Hyla arborea-group: (1) the group entered Eurasia from the east via Beringia from North America, during the Paleogene; and (2) the ancestors of the group reached Europe during the Early Miocene via the Tugai Strait between Europe and Asia (the Turgai Strait) and diversified, apparently, in Western Siberian. The Late Miocene and Early Pliocene records represent the most eastern expansion of the European genera, when the climatic conditions were still favourable for their distribution; it is conceivable for us that the Hyla savignyi may have potentially so far not found fossil occurrences in the Miocene of Eastern Europe and/or from the Caucasus in the south.

Bufonidae

Two groups of toads were found in the studied localities in Western Siberia; namely the common (Bufo bufo) and the green toads (Bufotes cf. viridis) (Figs. 7F–7K; Table S1). The toads of both groups, with their occurrences, are the most abundant among frogs remains found at the fossil localities.

Common toads

Bufo bufo is the recent species with the widest distribution (i.e. Central, Southern, Eastern Europe, and Western and Eastern Asia) of all members of the common toads Bufo bufo species group. This group includes three other species with limited distribution, namely: Bufo spinosus (Northern Africa, Western Europe), Bufo eichwaldi (south coast of the Caspian Sea), and Bufo verrucosissimus (east of the Black Sea) (Arntzen et al., 2013). These species are known also as the western group of the genus. Their nearby Eastern Asian relatives—the eastern group, include the Bufo gargarizans species group. The Western Siberian fossil record of the Bufo bufo species group is restricted to the late Late Miocene to the early Early Pliocene, which in comparison to the European record, is very poorly represented. The oldest toad remains that are assigned to the Bufo bufo species group are from the Middle Miocene of Slovakia: Bufo bufo from the Devinská Nová Ves—Zapfe’s fissure locality, 13.7–14 Ma (Hodrova, 1980; Böhme, 2003) and Bufo cf. bufo from the Devinská Nová Ves—Bonanza locality, 13.5–13.7 Ma (Hodrová, 1988). Then, since 9.2 Ma during the Late Miocene (Suchomasty locality in the Czech Republic) (Hodrová, 1987), Bufo bufo representatives are present in Central Europe and extend their distribution across Europe. At ca. 4.7 Ma, remains of the common toad, exhibiting characters of the Recent Bufo spinosus, appeared in Spain, in the Celadas 6 locality (Böhme & Ilg, 2003). The oldest fossil remains referred to Bufo verrucosissimus were recovered from the Late Pliocene (3.0–3.8 Ma) in the Apastovo locality in Russia (Ratnikov, 2001). The Western Siberian record suggests at least a late Miocene dispersal of Bufo bufo to the east, reaching the present distribution area of the species. Considering the genomic data of Recuero et al. (2012), these ‘migrants’ should represent the common ancestor of the Bufo bufo + Bufo verrucosissimus clade, expanding to the east into Asia and to the south into Eastern Europe. This bufonids most probably remained, permanently, in these areas, until present times. The lack of their representation in the fossil record in the Late Pliocene and Quaternary sites can be explained by sampling bias. Although Bufo bufo and Bufo verrucosissimus do not occur sympatrically nowadays, specimens of both these species have been found together in two Middle Pleistocene localities (Koziy Ovrag and Yablonovets from Russia; see more in Table S5).

Two recent molecular studies (Garcia-Porta et al., 2012; Recuero et al., 2012; pp. 71–86) suggested models of palaeobiogeographic history and timing of major cladogenetic events in the Bufo bufo species group; e.g. the origin in Southwestern Asia and subsequent migration into Europe via Anatolia. These studies, however, did not consider the entire fossil record, including the oldest record of the groups from the Middle Miocene of Slovakia (Hodrova, 1980) nor those of the species group in both their calibration of the molecular clock and palaeogeographic considerations. The updating and improvement of the distribution models are, therefore, necessary. Moreover, further finds of the fossil forms of southeastern species Bufo eichwaldi will help to reveal the place of origin and distribution routes of the ancestors of the group. Although only the molecular clock, and not the entire fossil record of the group has been used for the calibration, results from mtDNA sequencing seem to provide reliable data on diversification rates within the Bufo bufo species group, which can be confirmed by first appearances of the fossils related to each recent species.

Green toads

The range of the widely distributed Bufotes viridis species group (or Bufotes viridis sensu lato) extends across Central Europe to Central Asia, as well as the entire Northern Africa and Mediterranean area, including numerous islands. The species complex is highly diverse and includes over ten recognised species, e.g. Bufotes balearicus (Southern Mediterranean and Apennine Peninsula, Corsica, Sardinia, Balearic Islands), Bufotes boulengeri (Northern Africa), Bufotes siculus (Sicily), Bufotes viridis (Central and Eastern Europe), Bufotes variabilis (Balkans, Anatolia, Caucasus), etc., found in a diverse range of environments (Stöck et al., 2006; Stöck et al., 2008b). Among them, however, no valuable osteological characters have been established for taxonomic identification (Blain, Gibert & Ferràndez-Cañadell, 2010). Hence, no precise specific assignment of any fossil material is possible. Blain, Gibert & Ferràndez-Cañadell (2010) recently showed that the green toads were also present in the Iberian Peninsula in the Early Pleistocene, 1.1–1.3 Ma, and suggested that they became extinct due to climatic changes and/or competition.

In the studied Western Siberian localities, fossil remains that are related to Bufotes viridis are the most frequently occurring element in the Western Siberian herpetofauna. This species is almost permanently present from the Middle Miocene to Early Pleistocene. Specimens are found in the late Middle Miocene localities, and although there are gaps in the record, remains are present in the late Late Miocene to Early Pleistocene localities (Table S1). In the youngest localities (Olkhovka 1A, Lebiazhie 1A, Lebiazhie 1B), they are found as a sole taxon. Further fossils assigned to the family Bufonidae (Bufonidae indet.) were already reported from the Kentyubek locality in the Turgay Basin, from the Middle Miocene (Bendukidze & Chkhikvadze, 1976), and from two localities in the Zaisan Basin: the Zmei Gorynych locality in Akzhar Svita, from the Early Miocene (Chkhikvadze, 1985) and from the early Rupelian age fossil sites (see section ‘Pelobatidae’) of the Buran Svita (Chkhikvadze, 1998). Malakhov (2005) described the stratigraphically oldest green toad fossil, Bufotes aff. viridis, from the early Early Miocene (20.4–22.5 Ma, Aquitanian) locality of Ayakoz in Northeastern Kazakhstan (Fig. 1; Table S1). Bufotes aff. viridis from the Ayakoz locality is older than the Bufotes aff. viridis from the Early Miocene Keseköy locality (18–20 Ma) in Northwestern Turkey (Claessens, 1997). All the occurrences of the oldest European fossils of green toads are from the Early Miocene: Vieux-Collonges locality in France (14–17 Ma), (Bailon & Hossini, 1990); Petersbuch 2 and 7 (17.5–18 Ma) localities in Germany (Böhme & Ilg, 2003); and probably the Córcoles locality (17–18 Ma) in Spain (Sanchíz, 1998a). Once the green toads entered Europe, they became a regular element of the European Neogene and Quaternary herpetofaunal assemblages (Fig. 9). Besides Bufotes aff. viridis, the European record of green toads includes another species, Bufotes priscus, from four localities of the latest Early Miocene to the earliest Late Miocene age (see Table S5). Taking into account the Bufotes viridis Neogene records and the bufonid records from the Eurasian Paleogene, we suggest that the group arrived in the Old World in the Paleocene (Rage, 2003), entered Central Asia in the Early Oligocene and diversified there. Although we were not able to study the Paleogene bufonid record from Kazakhstan, taking into consideration the palaeogeography of common and green frogs, the assignment of the Early Oligocene Kazakhstan record to the green toad seems most probable. Apparently, the Early Oligocene forms were ancestral to the Bufotes viridis lineage, which evolved in Central Asia in the Early Miocene. This assumption is also supported by molecular data suggesting that: (1) the green toad clade underwent diversification in Asia during the Oligocene/Early Miocene; and (2) a high genomic and specific diversity is found within the Central Asian green toads (Stöck et al., 2006). Present in the Central Asian fossil record from the Early Miocene; they consequently dispersed via Anatolia in the Early Burdigalian into Europe during the Middle Burdigalian. Apparently, the European Neogene record should not necessarily represent one ‘lineage’ or one dispersal event of the Bufotes viridis group from Asia. Several migration events most probably took place during the Miocene. The descendants of these events were replaced later by the ancestors of the recent species Bufotes viridis, Bufotes variabilis, etc. as indicated by the genetic data at the Mio–Pliocene transition (Stöck et al., 2006). Prospective further studies could include: (1) the verification of dispersal events in the European fossil record, with help of an abundant and species-rich fossil material from stratigraphically well-dated localities; (2) the exploration the Miocene record of Anatolian and Southeastern Europe, as well as the Paleogene record of Asia; and (3) a challenging project of establishing the osteological characters that are important for the systematic identification of the members of the Bufotes viridis species group.

Ranidae

The family of true frogs, Ranidae, is present in the Western Siberian record by both green (Pelophylax sp.) and brown (Rana sp.) frogs. The green frogs appear more frequently in the record than the brown frogs. Both frog genera are common amphibians in the recent herpetofauna of the area. Besides this record, further true frog finds (e.g. Ranidae indet.) are reported from the early Rupelian age fossil sites (see the list of the locality section ‘Pelobatidae’) of the Buran Svita, Zaisan Basin. We were not able to revise their taxonomic validity due to lack of figured fossils and the inaccessibility of the material.

Green frogs

The genus Rana includes 21 recent species of aquatic frogs having a wide distribution ranging from Northern Africa, Europe to Eastern Asia. Two genetically distinct clades, i.e. Western Palaearctic and the Far East, are recognised within the green frogs genus Pelophylax (Lymberakis et al., 2007). The oldest green frog record from Western Siberia (Pelophylax sp.) is dated back to the late Middle Miocene, coinciding stratigraphically with the Eastern Siberian record of the group (Middle Miocene, ca. 13 Ma, Tagay Section, Baikal Lake, Russia) (Daxner-Höck et al., 2013). Records of this group are present in the studied localities until the late Early Pliocene with long (during the Late Miocene) and short (during the Early Pliocene) gaps in the fossil record. Due to the fragmentary preservation of the studied bones as well as the lack of other informative elements of the skeleton (e.g. frontoparietals), any assignment to the recent green frog species was impossible. Considering the present distribution of the two green frog clades, an affiliation of the Western Siberian fossil record to the Western Palaearctic clade is most probable.

Despite there being only a few green frog records described in this study, these records still significantly expand the previously scarce and poorly known fossil history of the genus. Moreover, both of the Middle Miocene records from Western and Eastern Siberia represent the oldest records of the green frogs in the Asian continent. Although an Asiatic origin of the green frogs has been already assumed by several authors, e.g. Sanchíz, Schleich & Esteban (1993) and Lymberakis et al. (2007), the earliest frog remains have been assigned to the Pelophylax ridibundus species group, which occurred in Europe in the Early Oligocene (Möhren 13 locality, Germany) (Sanchíz, Schleich & Esteban, 1993). Its affiliation to a living species is impossible. In Europe, the fossil record of Pelophylax is continuous and is maintained through the Oligocene and entire Neogene (Table S5). Nevertheless, a well-documented Paleogene record of the group is not available from Asia and, therefore, any interpretations would not be confident. The only possible scenario, taking into account both the fossil record and genomic data, is that the Western Palaearctic green frogs split from their Far East sister clade during the Eocene; they diversified in the territory of Europe and/or Western Asia during the Oligocene; they dispersed back to the East in the Middle Miocene; and eventually reached the territory of the Western Siberia.

Brown frogs

The genus Rana (subgenus Rana sensu Veith, Kosuch & Vences, 2003) is comprised of more than 15 species that are distributed throughout Eurasia. Similar to green frogs, there are two known lineages from the brown frog species, namely: the Western and the Eastern Palaearctic lineages (Veith, Kosuch & Vences, 2003). Based on the osteological characters, the studied Western Siberian brown frog remains show a relation to the Western Asiatic lineage of the genus Rana, more precisely to the Rana temporaria species group (sensu Veith, Kosuch & Vences, 2003). Among the late Paleogene and Early Miocene fossil frogs (Böhme & Ilg, 2003), in which the generic identification is unclear (Rana vel Pelophylax), only the frog remains from the Early Miocene in Dietrichsberg, Germany (Böhme, 2001) have definitely been assigned to the brown frog Rana cf. temporaria, representing the oldest known record of the group so far. As already suggested by Böhme (2001), brown frogs migrated from their possible centre of origin in Western or Central Asia to Europe during the second half of the Early Miocene. This hypothesis is confirmed by the brown frog fossils from the Ayakoz locality in Kazakhstan, which dates back to the Aquitanian age and are stratigraphically older than the Dietrichsberg fossil frogs. The present-day biogeography and diversity of brown frogs, the presence of a distinct Eastern Palaearctic lineage in Eastern Asia as well as the Asian distribution of many European species provide further support for an Asiatic origin. Most likely, the dispersal route of the brown frogs was similar to that of the green toad (Bufotes cf. viridis) whereby dispersal into Europe occurred via Anatolia, during the Early Miocene.

It is interesting to note that the earliest brown frog from the studied Western Siberian localities (Malyi Kalkaman 2) shows osteological similarities with the recent species Rana temporaria, representing herewith the oldest fossil record of the species in the east.

Previous molecular studies (Veith, Kosuch & Vences, 2003; Lymberakis et al., 2007), on both green and brown frogs, aimed to reconstruct their phylogenetic relationships, suggest models of biogeographic history as well as suggest when the splits between different genera, clades, species, etc. occurred. Such studies have provided contradictory results also for this group, e.g. the split of Rana and Pelophylax was at 9.32 Ma (Veith, Kosuch & Vences, 2003), whereas Lymberakis et al. (2007) estimated the split of the Western Palaearctic and Far East lineages of Pelophylax to have occurred significantly earlier, i.e. 15 Ma before. Here, neither geologic events nor the fossil records have been used consistently for the calibration of the molecular clock. Thus, the recalibrating of the timing for the splits with the new fossil finds provides a more reliable basis for phylogenetic reconstructions.

For the better understanding of relationships between these groups, as well as to reveal more around the origin and palaeobiogeographic history of them, it would be interesting to review the specimens of the Paleocene frogs (Ranidae indet.) from the early Rupelian fossil sites (see section ‘Pelobatidae’) of the Buran Svita in the Zaisan Basin (Chkhikvadze, 1998). The incorporation of such a review, however, was not possible in the present study, due to the lack of figures of the fossils and the inaccessibility of the material.

Gekkonidae

The family Gekkonidae is represented in the Western Siberian fossil record by the straight-fingered or even-fingered geckos, genus Alsophylax. They occur only in the Cherlak locality, dated back to the terminal Miocene, ca. 5.9 Ma. Alsophylax sp. is the most abundant element in the herpetofaunal assemblage of the Cherlak locality, with approximately 70% of the identifiable bone material belonging to this taxon. The genus Alsophylax is mainly distributed in Central Asia, partly occurring also in Mongolia and China. These geckos prefer habitats in arid and warm landscapes (Ananjeva et al., 2006). The appearance of these dry and warm adapted geckos in Western Siberia, which is 4°N of their present occurrence, suggests a shift of the arid environment from the south to the north at the end of the Late Miocene (see below). It is interesting to note that out of the seven gecko genera, e.g. Eublephareus, Mediadactylus, Terratoscincus (Ananjeva et al., 2006) inhabiting Central Asia, only Alsophylax, which has the most northern distribution, occurs in the fossil record. Apparently, this genus is ecologically more adaptable in comparison to other genera, not only in the present, but probably also in the past.

Lacertidae

Lacertid remains are the most frequent fossil bones among those of lizards occurring in Western Siberian localities. They are very rare in the Middle Miocene faunas, but occur more frequently in the Late Miocene, Pliocene, and Pleistocene localities. In the middle Late Miocene locality Pavlodar 1A (ca. 7.25 Ma), two taxa (Lacerta s.l. sp. 1 and sp. 2) occur sympatrically. Eremias sp. appears in the Western Siberian record in the Pliocene. This genus is widely distributed in the Central Asian steppes, inhabiting dry and warm habitats (Ananjeva et al., 2006).

Emydidae

Emydoidea sp. is the only turtle identified from the studied fossil sites. The present-day distribution of the monotypic genus Emydoidea is restricted to the water bodies of the northeastern territory of the USA. In Eurasia, fossil forms of this aquatic genus appear in the fossil record in Central Kazakhstan since the Middle Miocene (Emydoidea tasbaka, the Kentyubek locality in the Turgay Basin) (Chkhikvadze, 1989). Fossil forms have also been reported in Eastern Europe from the Late Miocene (Emydoidea tarashchuki, Krivoy Rog locality in Ukraine and Pantishara (8.7–9.2 Ma) in Georgia) (Chkhikvadze, 1980; Chkhikvadze, 2003). The Siberian record indicates their occurrence in Asia also during the Late Miocene, which, interestingly, is located much further north than their Middle Miocene record from Kazakhstan. According to Chkhikvadze (2003), representatives may have also been present in Eastern Europe during the Pliocene. We avoid interpreting palaeobiogeography, stratigraphic distribution, etc. of this genus, since the available published material (e.g. Chkhikvadze, 1983, 1989), together with other extinct testudinoid taxa from Kazakhstan and Eastern Europe, is insufficiently described and poorly illustrated, requiring thorough revision. Nevertheless, we used the available published data on both freshwater turtles and terrestrial tortoises to attempt to interpret the record at the family level (Table 2). The turtle records from three well-explored regions in the studying area, i.e. Zaisan Basin, Turgay Basins, and Western Siberia, are summarised in Table 2. Throughout the entire Early Miocene in the Zaisan Basin, the turtle fauna is dominated by aquatic forms, i.e. out of eight taxa only two are tortoises (Protestudo spp.). The aquatic forms remained dominant in the Zaisan Basin during the Middle Miocene, the terrestrial family Testudinidae completely replaced the aquatic turtles (Emydidae, Triochynidae) in the end of the Middle Miocene and became the only family present in the younger deposits of the Late Miocene. Similar to the Zaisan Basin, the aquatic forms represent the Middle Miocene turtle fauna in two adjacent regions, the Turgay Basin in the west and Western Siberia in the north. Subsequently, in the beginning of the early Late Miocene, a testudinid appears in Western Siberia and is replaced by an emydid towards the end of the late Late Miocene and a chelydrid at the Mio–Pliocene transition. The absence of tortoises since the end of the Late Miocene in Western Siberia and the Plio–Pleistocene in the Zaisan Basin can be explained by a less favourable, probably colder (MAT <15 °C, CMT <8 °C) climate. Since the late Late Miocene, the emydid and chelydrid aquatic turtles are the only chelonids in Western Siberia. The presence of these chelonids not only indicates a humid environment with standing water-bodies, but most probably also a cooler climate (for emydids: MAT >8 °C, CMT >−1.4 °C), since, in general, aquatic turtles can tolerate much colder conditions than tortoises, in that an aquatic environment acts as thermal buffer, consequently enabling aquatic turtles to populate higher poleward latitudes.

Table 2 Neogene testudinoid fauna of Western Siberia and the Zaisan and Turgay basins.

Stage		Zaisan Basin	Turgay Basin	Western Siberia	
	Svita	Turtle ‘Stage’*−	Taxa	Taxa	Taxa	
Pliocene					Chelydropsis kuznetsovi (Cy), ?Sakya sp. (Ey)	
Miocene	Late	Karabulak		*Protestudo illiberalis (Ts)		OEmydoidea sp. (Ey)	
	Kalmakpai		Protestudo kegenica (Ts)	KProtestudo karabastusica (Ts)	
Middle	Sarybulak	Up	*Protestudo darewskii (Ts)	***Chrysemys sp. (Ey), ?Ocadia sp. (Gey), Emydoidea tasbaka (Ey), Kazakhemys zaisanensis (Pl), ?Chelydropsis sp. (Cy)	*+Chrysemy sp. (Ey), Ocadia sp. (Gey)	
	Low	*Pelodiscus jakhimovitchae (Ty)	
Zaisan	Up	**Baicalemys moschifera (Ey)		
		Low	**Baicalemys sp. (Ey)	
Early	Akzhar	Up	Protestudo sp. (Ts)			
Middle	*−Chelydropsis poena (Cy)	
*Pelodiscus sp. (Ty)	
*−Kazakhemys zaisanensis (Pl)	
**Baicalemys jegalloi (Ey)	
	**Ocadia iliensis (Gey)	
Low	*−Protestudo alba (Ts)	
Emydidae gen. indet. (Ey)	
Notes:

The data are summerized following to Chkhikvadze (1989), as well as the superscriptions before the taxa indicate the reference:

* Kordikova (1994);

** Danilov, Cherepanov & Vitek (2013);

*** Kentyubek fauna (Supplemental Information 3);

*− Chkhikvadze (1989);

*+ Tleuberdina et al. (1993);

K Kuznetsov (1982);

O our results.

The aquatic families are indicated in blue and terrestrial families in dark yellow colour. Ty, Trionychidae; Cy, Chelydridae; Pl, Platysternidae; Ts, Testudinidae; Ey, Emydidae; Gey, Geoemydidae.

Palaeobiogeographic considerations

By comparing the spatial and temporal patterns between European and Asian fossil records, including the first and last fossil occurrences, combined with an analysis of the available genomic data of the recent relatives of the fossil groups present in the studied material, certain palaeogeographic distribution patterns can be revealed along with new interpretations.

Our analysis suggests a Western Asiatic origin for Hynobiidae, Proteidae, aff. Tylototriton, Bufotes viridis species group and brown frogs, Rana. The green toads and brown frogs dispersed coincidentally in the earliest Miocene wherein, and at least for the Bufotes viridis group, Anatolia was involved. Anatolia also played an important role in the distribution of the Bufo bufo species group; however, any age estimation of the event is not available. A salamander, showing affinities to the clade of the recent East Asian genera Tylototriton + Echinotriton, is present in Western Siberia, most probably representing the forms similar to that of the Early Oligocene (aff. Tylototriton) in Europe, a sister group of the recent clade. In order to resolve the affiliations of these fossils, further Paleogene materials from both the Asia and European continents are necessary.

An eastward dispersal from Europe into Western Asia can be observed over a period ranging from the Middle to Late Miocene, based on the current data available from both European and Asiatic records, for at least seven amphibian groups (family Palaeobatrachidae, genera Chelotriton, Pelobates, Bombina (i.e. Bombina (cf.) bombina), Hyla (i.e. Hyla cf. savignyi), Pelophylax?, Bufo bufo species group). Besides the amphibians, some Western Siberian reptiles, such as the glass lizards from the Middle Miocene, show European affinities, resembling the Central European faunas (Vasilyan, Böhme & Klembara, 2016).

The amphibian genera Bombina, Hyla, Bufo, Rana and Pelophylax resemble a comparable palaeobiogeographic pattern: the molecular genetic data showed the presence of two clearly separable western and eastern clades (species groups) in each of these genera. In all cases, it was possible to morphologically attribute the Western Siberian fossil amphibians to the western clades or species of the clades. It is interesting to note that even though the first fossil occurrences of these genera have different stratigraphic ages, they are found exclusively in Europe (see Fig. 9; Table S5). To explain this common pattern, we hypothesise that the western and eastern clades had already split in the Paleogene, most probably in the western or central parts of Asia, and subsequently dispersed into Europe.

The Western Siberian fossil Mioproteus, Chelotriton, Bombina, Paleobatidae, Hyla, Bufo bufo and Rana temporaria represent the most eastern records of those groups found in the Eurasian fossil record. In comparison to their present-day geography, the Western Eurasian species of the genera Bombina and Hyla, respectively, show wider distribution ranges during the Middle to Late Miocene, and Late Miocene to Early Pliocene. The palaeogeographic affinity of the earliest Messinian pelobatid (locality Selety 1A) is still unclear. Considering the geographic location of the fossil site, its relation to the recent genus Pelobates seems most possible.

In Chkhikvadze (1985), two lizards Varanus sp. and Agamidae indet. have been reported from three Miocene localities of the Zaisan Basin. Although the taxonomic assignment of the remains could not be verified in this study, we adopt the identifications for biogeographic and palaeoenvironmental interpretations. These lizards are currently widely distributed in Central Asia. Varanus, being a thermophilous reptile species, is restricted to the southern part of the region. Its presence in the early Late Miocene of the Zaisan Basin aids in characterisation of the climate of the Sarybulak Svita, in the beginning of the Late Miocene, i.e. a probable MAT of not less than 14.8 °C (Böhme, 2003).

In summary, Western Siberia (Central Asia) can be hypothesised as a centre of evolution and dispersal for several temperate Neogene herpetofaunal taxa, e.g. the genera Salamandrella and Mioproteus, the green toad Bufotes viridis species group and brown frog Rana. The Neogene herpetofauna of Western Siberia and the adjacent areas has significant similarities with the European amphibian and reptile assemblages. The Western Palaearctic herpetofauna gradually entered the Siberian territory from Europe, between the Middle Miocene to Early Pliocene, strongly shaping the herpetofauna of Western Siberia and partially retaining the faunal elements of an Asiatic origin (e.g. Hynobiidae, Proteidae, and Alsophylax). The faunal diversity of the fossil record collapses significantly after the Early Pliocene. Only a few amphibians and reptiles, e.g. Salamandrella, Bufotes, Lacerta, and Vipera are present in the Pliocene fossil record, being able to survive in the increasingly less favourable environments to form the main part of the present-day Western Siberian herpetofauna.

The palaeobiogeographic analysis of the recent amphibian faunas of Western Asia (Savage, 1973; Garcia-Porta et al., 2012) hypothesised a progressive aridification of Central Asia linked with the global cooling trends during the Miocene, forcing amphibians to shift their distribution to the south.

Palaeoclimatic implications

The Neogene climate evolution of Western Siberia has been previously reconstructed based on palynofloras, showing a progressive change in environmental conditions, i.e. in the climate and vegetation, during the Miocene (Arkhipov et al., 2005). Between the Early to Late Miocene, a warm and humid climate was replaced by a warm temperate climate in the Middle Miocene and a boreal-warm temperate climate in the Late Miocene. Towards the end of the Miocene, a drastic climatic shift took place resulting in semiarid and arid conditions. The Pliocene climate is predominated by frequent changes between semiarid forest-steppe/steppe and arid desert environments, however, from the Late Pliocene the environment changes into subarctic (Arkhipov et al., 2005; p. 76, Fig. 46).

At a lower temporal resolution, the testudinoid fossil records from the Zaisan Basin, the Turgay Basin, and Western Siberia confirm a general trend towards aridity in the Neogene (Data S4). Based on the environmental requirement (aquatic or terrestrial) of the testudinoids from the Zaisan Basin, we infer that the climate changed from humid to dry. We further infer that the Early and Middle Miocene was mostly humid (dominance of aquatic families), whereas the presence of exclusively terrestrial forms (tortoises) from the latest Middle Miocene to Late Miocene indicates dry and open habitats in the Zaisan Basin. Unfortunately, it is impossible to make any quantification of the palaeoprecipitation values based on these limited taxa and well-documented herpetofaunal assemblages are necessary from these deposits for further environmental reconstructions.

To establish a better palaeoclimatic understanding, we estimated palaeoprecipitation values for 12 data points (Table S4). These localities provided six and more amphibian and reptile taxa, applicable for the bioclimatic analysis (Böhme et al., 2006). Even so, our data do not enable accurate reconstruction of the climate development over the Middle Miocene to earliest Pleistocene in Western Siberia. The climate development can, therefore, only be reconstructed and discussed for several short intervals. Nevertheless, our estimations rather show a dynamic climate development in the Neogene of Western Siberia, with larger precipitation amplitudes, ranging from 158 mm to over 1,500 mm per year (Table S1; Fig. 10), than previously estimated using palynological data (Arkhipov et al., 2005). Apart from the fluctuating humidity factor, in general, the MAP was significantly above the present day values (reaching 550% of the present-day values) (Fig. 10). Only two localities are characterised by drier climates, the late Serravallian (ca. 12.1 Ma) and the late Messinian (5.9 Ma), exhibiting either present-day or below present-day levels.

Figure 10 Palaeoprecipitation development of Western Siberia including the Zaisan Basin.

(A) Curve displaying the development of the absolute values of mean annual precipitation (MAP); (B) the ratio of MAP to recent precipitation value (MAP/MAPrecent100%), dashed black line (100%) indicates the recent precipitation values. Localities: 1, Ayakoz; 2, Vympel; 3, Poltinik; 4, Tri Bogatyrja; 5, Kentyubek; 6, Malyi Kalkaman 2; 7, Malyi Kalkaman 1; 8, Baikadam; 9, Novaya Stanitsa 1A; 10, Cherlak; 11, Detskaya zheleznaya doroga; 12, Olkhovka 1B.

Reliability of precipitation estimates

The accuracy of precipitation estimates, based on bioclimatic analysis of herpetofauna, depends primarily on the taxon counts and the assumption of low (stochastic) taphonomic bias (Böhme et al., 2006). In Western Siberia, some of the documented localities were rich in aquatic herpetofauna, e.g. composed by freshwater turtles, giant salamanders, proteids, etc., but small terrestrial forms (e.g. lizards and anguids) were absent, indicating a possible non-stochastic taphonomic bias (i.e. exclusion of elements of certain habitats). These localities will result in a bias in humidity estimates towards the wet end. Examples of such localities include Kentyubek and Novaya Stanitsa 1A, where the numeric results well exceed the MAP of 1,600 mm, the upper limit to which the eco-physiologic index—humidity relation is calibrated (see details in Böhme et al., 2006). In these cases, we restrict our estimates to a limit of 1,500 mm.

Aquitanian

For the Aquitanian age Ayakoz locality, we estimated a MAP value of 945 mm, representing more than three times higher rainfall in comparison to the recent times. Using the palynologic data, Arkhipov et al. (2005) estimated a humid climate with MAP 800 mm for the Abrosimov Svita (Aquitanian age) in Western Siberia. Besides this study and based on the data of fossil macroflora, Bruch & Zhilin (2007) estimated similar values of precipitation (935–1,232 mm) for about 30 Aquitanian age localities, distributed from Western to Eastern Kazakhstan. Our reconstruction, therefore, appears to fit well within the historical precipitation estimates of the region.

Akzhar Svita

Towards the end of the late Early Miocene (Burdigalian), an elevated humidity in Western Siberia can be suggested based on the presence of the giant salamander in three localities of the Zaisan Basin (Tri Bogatyrya, Vympel, and Poltinik). As already suggested, their occurrence indicates a high rainfall for those time periods (MAP >900 mm), as well as an increased basinal relief enabling the distribution and reproduction of this group in the lowland settings (Böhme, Vasilyan & Winklhofer, 2012). This period of the Akzhar Svita also corresponds to the folding and uplift of the Altai Mountains (Zykin, 2012; p. 394), from which the establishment of the higher basinal relief was possible.

Late Serravallian

In contrast to the already known climate development suggested by Arkhipov et al. (2005), our data suggest that there were strong humidity fluctuations during the late Middle Miocene (late Serravallian), with MAP values ranging between 282, 884 and 1,108 mm (Fig. 10). The only botanical data of this time (Bescheul macroflora) point to a warm-temperate and humid (MAP ∼700 mm) climate (Arkhipov et al., 2005), which best compares to our Malyi Kalkaman 2 results (MAP 884 mm).

Novastanitsa Svita

Although the herpetofaunal assemblage for the early Messinian locality Novaya Stanitsa 1A is incomplete, a very high MAP value of at least 1500 mm can be estimated. The value indicates a significantly higher humidity than of Tortonian–Messinian boundary and late Messinian (see below). Our data are contrary to the palynologic results, which gave lower estimates (400–450 mm; Arkhipov et al., 2005).

Rytov Svita

The Cherlak locality (5.9 Ma, Rytov Suite) is characterised by a rather dry climate (MAP 255 mm), with a similar humidity level to that of the present-day (Fig. 10). Our data for a warm and dry climate are confirmed by the presence of: (1) gekkonid Alsophylax; (2) mollusc fauna containing thermophilous species; (3) the small mammal fauna, represented mainly by pikas, hamsters, and jerboas, characteristic for open and dry habitats (Zykin, 2012); and (4) ostriches (Struthiolithus sp.) and camels (Paracamelus sp.) in this svita (Shpanskiy, 2008). Arkhipov et al. (2005) summarised the available palynological and vegetation data of the svita and reported the presence of a poor (due to the oxidation) spectra containing xerophyte plants (Asteraceae, Chenopodiacea), characterising desert and steppe environments. Interestingly, his results proposed a northward shift of dry steppe and desert environments by 4° (to the latitude of 56°), which concurs with our data, as is indicated by the presence of the steppe-dwelling gekkonid Alsophylax sp. (see section ‘Gekkonidae’).

Miocene–Pliocene transition (Detskaya Zhelznaja Daroga)

Even though the precise taxonomic identification of the Western Siberian and Zaisan cryptobranchids, is unclear at the generic or species level, their occurrence indicates a high rainfall >900 mm MAP (Böhme, Vasilyan & Winklhofer, 2012) during the Burdigalian age in the Zaisan Basin and the Miocene–Pliocene transition in Western Siberia. Besides the presence of Cryptobranchidae indet. from the locality Detskaya Zheleznaya Doroga, the co-occurrence of the aquatic chelonids Chelydropsis kuznetsovi and probable Sakya sp. (Gaiduchenko, 1984; Gaiduchenko & Chkhikvadze, 1985) confirms the presence of a high degree of precipitation at the Miocene–Pliocene boundary in Western Siberia.

Earliest Pliocene (Olkhovka 1A–1C)

Our earliest Pliocene humidity data are estimated based on the fauna from the localities Olkhovka 1A, 1B, and 1C, for which no correlation data is available for regional svitas (see section ‘Geology and Stratigraphy’). Nevertheless, the results still indicate significant precipitation (MAP 575 mm), well above the present-day values for this region. These findings correspond well with the similar aged Speranovskaya palynoflora (Volkova, 1984), which indicates the presence of warm forests and forest-steppes with MAP estimates between 500 and 550 mm (Arkhipov et al., 2005).

Conclusion

In summary, over 50 salamander, frog, lizard, snake, and turtle taxa have been assigned to specimens from more than 40 Western Siberian localities that range in age from the Middle Miocene to the Pleistocene (Table S1). The late Middle Miocene localities have the most diverse faunas including all the main groups of the herpetofauna. According to our analysis, the fossil fauna contains taxa showing an Asian (Eastern Palaearctic) origin, such as Hynobiidae, Proteidae, Bufotes viridis species group and Rana, Varanus, and Agamidae. The main part of the herpetofaunal assemblage, including Palaeobatrachidae, Paleobatidae, the genera Chelotriton, Bombina (i.e. Bombina (cf.) bombina), Hyla (i.e. Hyla (cf.) savignyi), Pelophylax?, Bufo bufo, Ophisaurus sp. (Vasilyan, Böhme & Klembara, 2016), has European (Western Palaearctic) affinities. The Western Siberian records of Mioproteus, Chelotriton, Bombina, Paleobatidae, Hyla, Bufo bufo, and Rana temporaria represent the most eastern occurrences of these groups in Eurasia. The earliest Miocene dispersal of the green toad, Bufotes viridis species group into Europe from Asia via Anatolia, can be inferred. We suggest the same distribution pattern for brown frogs, Rana, too. In this scope, it will be important to perform future detailed studies on the Neogene record of the amphibian and reptile faunas in Anatolia and analyse them in a palaeobiogeographic context.

According to our study, the precipitation development in Western Siberia shows high-amplitude changes during the studied intervals. Aside from the certain time periods, i.e. late Serravallian and late Messinian, the palaeorainfall in Western Siberia was estimated to be significantly higher than the present-day values. The best results on precipitation estimates that we were able to reconstruct, with reliable age constrain, were for the period from 6.6 to ∼4.5 Ma. These results indicate a humid climate during the early Messinian; a dry climate during the late Messinian; a very humid climate during the Miocene–Pliocene transition and a humid climate during the earliest Pliocene (Data S4; Fig. 10). The decreasing tendency of the herpetofaunal diversity towards the end of the Neogene and Quaternary could be attributed to the progressive global cooling and forced ice-sheet development in the Northern Hemisphere.

Supplemental Information

Supplemental Information 1 Overview of the amphibian, reptile and small mammal faunas from the Western Siberian localities studied in this work (in red) and according to published data (in pink).

Asterisks indicate the localities according to the literature data, the material of which was partially studied in this work. The species of small mammal linages are indicated in similar colours. For details on published data see Supplemental Information 2. Abbreviations: akz, Akzhar; tr, Turme; zs, Zaisan; klm, Kalkaman; ish, Ishim; pv, Pavlodar; sb, Sarybulak; klp, Kalmalpai; kb, Karabulak Svita; kd, Kedey; nst, Novaya Stanitsa; rt, Rytov; is, Isakov; psh, Peshnev; krt, Krutogor; bt, Betekey; liv, Levetin; sel, Seletin; irt, Irtish; kar, Karagash; kc, Kochkov.

Click here for additional data file.

Supplemental Information 2 Ages, biochronologic, magneto- and lithostratigraphic data of the studied localities from Western Siberia.

The numbering of the localities corresponds to the locality numeration in the Figs. 1, 2, S1 and S4.

Click here for additional data file.

Supplemental Information 3 List of localities with amphibian and reptiles faunas according to the literature data, accompanied with other vertebrate groups.

We provide also available assignments to stratigraphic horizons, ages, svitas, basins and literature sources. The locality numbering corresponds to the locality enumeration given in the Figs. 1 and 2, as well as in the Supplemental Information 1.

Click here for additional data file.

Supplemental Information 4 Precipitation reconstructions.

Western Siberian localities, their ages, estimated palaeoprecipitation values, present-day precipitation near the localities and their reference climate station, precipitation difference to present day values, relative precipitation to present-day values, and faunal content.

Click here for additional data file.

Supplemental Information 5 Fossil records of the studied groups.

European fossil occurrences of Hynobiidae, Proteidae, Chelotriton, Tyltotriton, Bombina, Palaeobatrachidae, Pelobatidae, Hyla, Bufo bufo gr., Bufotes viridis gr., Rana, Pelophylax according to Böhme & Ilg (2003).

Click here for additional data file.

We sincerely thank B. Sanchiz (Madrid), Z. Roček (Prague), J. Prieto (Munich), M. Rabi (Tübingen), M. Delfino (Turin) and V. Ratnikov (Voronezh) for their constructive discussions and comments. We are grateful to V. Chkhikvadze (Tbilisi) for providing material from the localities: Pavlodar 1A, Ayakoz, Petropavlovsk 1/2, Malyi Kalkaman 1; Dr. L. Maul (Weimar) for providing details of the ages of the Quaternary localities where palaeobatrachid frogs occur; A. Fatz (Tübingen) for making figure and table images; I. Stepanyan (Yerevan) for literature help; and A. Ilg (Düsseldorf) for providing support with the database ‘fosFARbase’. Finally, we would like to express our gratitude to G. Piñeiro (Montevideo) for editorial comments and guiding, and both reviewers M. Venczel (Oradea) and Sh. Meiri (Tel Aviv) for their criticism and constructive reviews.

Institutional/collection Abbreviations

GPIT Paläontologische Sammlung der Universität Tübingen, Tübingen, Germany

HC Collection of Marcela Hodrova (Prague University), now stored in GPIT

MNCN Museo Nacional de Ciencias Naturales, Madrid, Spain

NMNHK National Museum of Natural History, Kiev, Ukraine

PIN Palaeontological Institute, Russian Academy of Sciences, Moscow, Russia

GNM National Museum of Georgia, Tbilisi, Georgia

GIN Geologic Institute, National Academy of Russia, Moscow, Russia.

Anatomical Abbreviations

ao antrum olfactorium

alo antrum pro lobo olfactorio

dl dental lamina

ds dental shelf

hl horizontal lamella

is incisura semielliptical

ff frontoparietal facet

fcpr facial process of maxilla

fMx5 foramina for mandibular division of the fifth cranial (trigeminal) nerve

hfr haemal foramen

hl horizontal lamella

lf lacrimal facet

lg longitudinal groove

lh lamina horizontalis

lp lateral processes

ls lamina supraorbitalis

mc Meckelian canal

na neural arch

nc neural canal

nf nasal facet

onf orbitonasal foramina

olf olfactory foramina

pf parasphenoid facet

pfc palatine facet

ph paries horizontalis

prz prezygapophysis

psz postzygapophysis

pv paries verticalis

pxp premaxillary process

pyp pterygapophysis

sac opening of superior alveolar canal

sg symphyseal groove

sf splenial facet

tpr transverse process.

Additional Information and Declarations

Competing Interests

Author Contributions

Data Deposition

1 In the town of Petropavlovsk, two fossil sites (Petropavlovsk 1 (MN12) and Petropavlovsk 2 (MN14)) having different ages are known, see Zykin (2012). Since the enclosed collection label to the material indicates only ‘locality Petropavlovsk, leg. 1972’ any stratigraphic allocation of the fossils to one of those layers is impossible.

2 We follow taxonomy suggested by Fritz, Schmidt & Ernst (2011) recognizing Emydoidea as a distinct genus from Emys.

3 Localities: Zertsalo (Sunduk Section), lager Biryukova (Kiin–Kerish Section), lower faunistic level of Plesh (Kusto–Kyzylkain Section), probably also Tabtym (Sarykamysh Section).

4 Localities: main level of Plesh, Tuzkabak, Cherepakhovoe Pole (Tayzhuzgen Section), Raskop (Aksyr Section), Tyubiteika, sopki ‘Rybnaya’, and Kontrolnaya (Juvan–Kara Section).

5 Localities: Maylibay, Tologay (Tayzhuzgen Section), and Podorozhnik (Jaman–Kara Section).

6 The divergence dates of split events were estimated by a relaxed molecular clock approach, based on the mitochondrial data set, where the calibration with fossil record is missing.

The authors declare that they have no competing interests.

Davit Vasilyan conceived and designed the experiments, performed the experiments, analysed the data, contributed reagents/materials/analysis tools, wrote the paper, prepared figures and/or tables, and reviewed drafts of the paper.

Vladimir S. Zazhigin contributed reagents/materials/analysis tools and reviewed drafts of the paper.

Madelaine Böhme conceived and designed the experiments, analysed the data, contributed reagents/materials/analysis tools, and reviewed drafts of the paper.

The following information was supplied regarding data availability:

The raw data is included in the figures.

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
