# Peer review of "Neogene amphibians and reptiles (Caudata, Anura, Gekkota, Lacertilia, and Testudines) from the south of Western Siberia, Russia, and Northeastern Kazakhstan"

_PeerJ, doi:10.7717/peerj.3025_

## Round 0.1 · original submission · Major Revisions

Dear authors,

I have now two reviews about your manuscript. They considered that the manuscript deserves to be published as it represents an important compilation of fossil records of reptiles and amphibians from a large area of Siberia and Central Asia. At least two big problems are easily visualized when reading the text: The English language and the paleoenvironmental analysis. You must revise both carefully, perhaps it can be optimum that you ask for the help of a natural English speaker. One of the reviewers reported problems with the assignation of the specimens in the text, which is not coincident with that you show in the figures and requested complementary descriptions for taxonomic assignations based on more diagnostic characters. So, please, make the arrangements needed and modify the text according with the required by the reviewers. Offer us more evidence that support your paleoenvironmental inferences, which are interesting but lack the source of confident, statistically evaluated data (if possible), for proving them. I hope to see an improved version of your manuscript very soon, in order to continue with the editorial process. In other case, please provide a rebuttal letter explaining your points of view.

With my best regards,
Graciela Piñeiro

·

Basic reporting

The English should be improved by a native English.

Experimental design

No comments

Validity of the findings

In one case I do not agree with the identification of the authors (namely Rana arvalis).

Additional comments

The manuscript "Neogene amphibians and reptiles (Caudata, Anura, Gekotta,
Lacertilia, Testudines) from south of Western Siberia, Russia
and Northeastern Kazakhstan" (#12606) is an important contribution on Neogene lissamphibians and reptiles from Western Asia and as such it fills a gap in our knowledge about faunal diversity, biogeograpgical links to European faunas and paleoenvironmental reconstruction.
The manuscript should be accepted after a minor revision of the anatomical descriptions and taxonomic identifications. The English should be also improved (I made several corrections but I am not a native English).
Main comments:
1. I agree with most assignments and descriptions, but sometimes the taxonomic part is not in agreement with the abstract (e.g. the authors talk about Hynobiidae in the abstract and about Salamandrella in the taxonomic part...), or the description and the illustrated bones point to another taxa [e.g. see line 721 and figure 6N, where the authors state that "the dorsal crest is low" and assign the specimen from Malyi Kalkaman 1 to R. arvalis - ... in R. arvalis the dorsal crest is high (in the latter taxon the dorsal crest it is of same height as the tuber superius...)]; see also comments within the MS - line 514.
2. Additional anatomical features should be used when describing the vertebrae of Hynobiidae (Salamandrella) and Mioproteus, as follows: presence or absence of spinal nerve foramina in the trunk and caudal vertebrae (at least in Salamandrella both vertebral regions are represented in the material); presence or absence of basapophyses (anterior or posterior).
3. Mioproteus is identified based on vertebrae and "maxillae" (lines 292-325). Since the maxilla in Proteus and presumably in Mioproteus is reduced the identity of the described specimens should be revised...
4. The vertebrae of Chelotriton (lines 367-388) are compared to many taxa (Recent and extinct). Here, comparison to Carpathotriton, which is also a fossil pleurodeline salamandrid, should be included.
5. The statement about the "Pliocene disappearance of Palaeobatrachidae from Central Europe" (lines 1106-1110) simply is not true and it should be revised [e.g. the type material of Palaeobatrachus (= Pliobatrachus) langhae is from Betfia 2 locality, which is of early Pleistocene age].
The annotated version of the MS (with comments and proposed corrections) is also included.
Márton Venczel

·

Basic reporting

Please fix the English, and carefully report he methods and data used in palaeo-climatic reconstruction, paying attention to reporting the places and times accurately, clearly writing the methods, and explaining the statistical procedures, if any

Experimental design

No Comments

Validity of the findings

I had little faith in the reconstructed values for climate (precipitation and temperature. Because data are presented: 1. Without defining a method to do so. 2. Without saying what data were used to do it in any particular case. 3. Without defining how values for recent taxa were calculated, and from where and 4. Without any indication of variability in the estimates (standard deviations?). Hence I have very little faith in the reconstructed values.

Additional comments

This is a potentially important manuscript that reviews, and adds much data on, fossil reptiles from a large region of Central Asia, during the Neogene. The dearth of such remains is a very good motivation for publishing such a paper and database. It is a very long paper, full of minute detail, that while useful, do not make for a very easy reading. I am not an expert (or even a novice) paleontologist, so I cannot comment on data quality or accuracy, and explicitly trust the authors to have done a good job.

That said, as someone more used to examining climatic models, I had little faith in the reconstructed values for climate (precipitation and temperature. Because data are presented: 1. Without defining a method to do so. 2. Without saying what data were used to do it in any particular case. 3. Without defining how values for recent taxa were calculated, and from where and 4. Without any indication of variability in the estimates (standard deviations?). Hence I have very little faith in the reconstructed values.
The other worrying aspect was the language. I spotted numerous issues with the English – basically a manuscript cannot be published with such poor attention to language issues. Please fix it professionally.
Aside from these I have but minor comments:
Title” perhaps south-western Siberia rather than “south OF western Siberia”?
Abstract, Methods and Results: replace “Russia” with “Russian”, replace “herpetofaunal assemblage” with “herpetofaunal assemblages”, and fix English and tense etc. (has and have and the like; the word “the” is missing from quite a few places etc., replace divers with diverse)
Abstract, Conclusions: all the conclusions refer to amphibians. What about reptiles?
Line 67: add the at the beginning of this sentence (THE Region with studied Neogene outcrops belong to…”)
Line 74: why “about”?
Line 80 “Amphibiaweb” is a poor source of geographic information. If the authors wish to use it I also suggest using more up to date versions than the 2012 one. Why not use the IUCN data?
“This is the poorest regional fauna of the Palaearctic Realm” really? What about the Sahara and arid parts of North Africa and Arabia? What about regions lying above the Arctic Circle? Please re-consider
Line 86: add “and “ before “Eremias arguta”
Line 91: change “widely” to “wide”
Line 130: what’s the “Zaisan Basin”?
Line 134: delete “the” or explain lists of what
Line 153: Eremias sp., and Coluber sp. Are reptiles, but the preceding text refers to amphibians only
Line 157: replace “Western Siberian” with “Western Siberia”
Line 163: I am sure they are not “numerous” – there are more in the present work, and they are numerated by the authors. Change to “many”?
Lines 169-171 “To avoid the confusion with the locality names, used by different authors in the Russian literature, here we provide all known names for the fossil localities as well.” – Excellent approach. I would suggest augmenting it with: 1. publishing the list as a table or a supplement, and 2. Give latitude and longitude values for each localities. Names and spellings change – latitudes and longitudes stay constant (continental drift notwithstanding…)
“collection numbers” – lines 179-191 – perhaps provide as an appendix? It hurts flow here. Just an idea
Lines 213-215: “the faunas of two and more localities with age differences less than 100 kyr and/or belong to a single stratigraphic unit – svita, are considered as one.” – change to two OR more. You give a temporal range (100kyr) – is there also a spatial range? Will you unite two localities if they are 100km apart? And 1000km? and 5000km?
Line 893: you had Lacerta s.l. sp. 1 and Lacerta s.l. sp. 2: why here is it just “Lacerta s.l. sp.” And not “Lacerta s.l. sp. 3”?
Lines 984-985: “the amphibians and reptile remains retained entirely unstudied.” – this is not entirely true for all of Asia. From my own country (Israel), for example, I know of at least one recent major fossil locality (Kessem Cave; see Maul, L. C., Bruch, A. A., Smith, K. T., Shenbrot, G., Barkai, R. and Gopher, A. 2015. Palaeoecological and biostratigraphical implications of the microvertebrates of Qesem Cave in Israel. Quaternary International and Maul, L. C., Smith, K. T., Barkai, R., Barash, A., Karkanas, P., Shahack-Gross, R. and Gopher, A. 2011. Microfaunal remains at Middle Pleistocene Qesem Cave, Israel: preliminary results on small vertebrates, environment and biostratigraphy. Journal of Human Evolution 60: 464-480.). The authors themselves refer to Anatolia (line 996) The statement needs to be softened a little.
Line 1003: capital A in Asiatic
Line 1075: replace “taxa” with “taxon”
Line 1407: provide reference for “this genus is widely distributed in the Central Asian steppes, inhabiting dry and warm habitats”
Line 1449: small r in recent please
Line 1457: “is” or “was”?
“an eastward dispersal from Europe into Western Asia can be observed over a period of time ranging from the Middle to Late Miocene.” – how do you know, given the dearth of Asian records alluded to earlier, that this is indeed a dispersal in this direction, rather than an artefact of not having found earlier remains in Asia because search effort was so much poorer – over a much larger region?
“molecular genetic data showed presence” – please provide references
Lines 1492-1493: “Western Siberia (Central Asia) can be hypothesized as a centre of evolution and dispersal for several temperate Neogene herpetofaunal taxa” – what is a centre of dispersal? And how does a centre of evolution sit with the notion that the taxa are in fact, of European origin (see above)?
Lines 1512-1514: what are the differences between “warm and
humid climate”, ”warm temperate” and ”boreal-warm temperate”? unclear
“mean annual precipitation (MAP) is significantly above the present day values (reaching 550% of the present-day values)” – where and when? Over such a vast array and immense period of time you need to be much, much more specific. Surely not for the entire region? And WHEN?
Lines 1547-1548: replace “is” and “are” with “were”

Shai Meiri

---

## Round 0.2 · Minor Revisions

Dear authors,

I am glad to see that you paid attention to all the reviewers’ recommendations, improving substantially the manuscript. However, the use of the English language and grammar are still very bad. I have modified the text, trying to improve a little this problem, but I am also not an English speaker. You may still want to consider further editing of the language. Please, correct carefully all the suggestions that I made using the tracking system of Word. If you disagree with any of them, you have to explain the reasons in a rebuttal letter.

You should also integrate figure references to the characters that you are describing, mainly to the vertebral and the cranial remains as I recommend in the tracking text as a commentary.

My recommendations for figure captions are in the annotated pdf uploaded along with the tracked changes word file.

Finally, please check the references and make sure that all of them are cited in the text and of course, that all citations in text are included in the reference list.

I very look forward to see the reviewed version of the manuscript with all the suggested modifications included.

With my best regards,
Graciela Piñeiro

---

## Round 0.3 · Minor Revisions

Dear authors,

I am glad to see how much improved your manuscript is. I have annotated just minor corrections in the file attached to this letter. I hope you find them useful and look forward to see your updated version submitted very soon.

With my best regards,
Graciela Piñeiro

---

## Round 0.4 · Minor Revisions

Dear Davit,

Thank you very much for your resonse letter. I will agree with your explanations about refusing some of the minor grammatical changes that I suggested. Your manuscript is a very important and useful compilation about Neogene amphibian and reptilian records of Russia, including detailed anatomical descriptions, taxonomic reviews and paleobiogeographic analyses. I am very glad to have handled this manuscript.

I found just a minor grammatical error in your figure captions: please, see the caption for figure 7 in the attached pdf and change ‘spleneal’ by ‘splenial’ and resubmit. I will be happy to accept the manuscript then.

Best regards,
Graciela Piñeiro

---

## Round 0.5 · accepted · Accept

Congratulations, your article is now ready for publication in PeerJ!

Best wishes,
Graciela Piñeiro